# Cert-LAS: Toward Certified Model Ownership Verification for Text-to-Image Diffusion Models via Layer-Adaptive Smoothing

Leyi Qi [1]  Yiming Li [1]  Siyuan Liang [1]  Zhengzhong Tu [2]  Dacheng Tao [1]

## Abstract

Large-scale text-to-image (T2I) diffusion models have enabled unprecedented creative applications, but their unauthorized use has raised serious intellectual property concerns, making model ownership verification (MOV) increasingly critical. We find that existing backdoor-based diffusion watermarking methods often (implicitly) assume a "faithful" verification process, namely, that the verifier can query a suspicious model and obtain the faithful watermark response to complete MOV. However, in practice, adversaries may intentionally or unintentionally damage potential watermark signals, significantly degrading verification reliability. To address this issue, we propose Cert-LAS, the first certified MOV method for T2I models based on layer-adaptive smoothing. In general, Cert-LAS embeds specified watermarks using diffusion classifiers and an LFS-guided layer-adaptive noise, and verifies ownership by examining whether the suspected model exhibits significantly stronger watermark responses compared to unwatermarked references through hypothesis testing. We further prove that, under certain conditions, our Cert-LAS can still achieve reliable verification even in the presence of malicious removal attacks. Extensive experiments validate the effectiveness of Cert-LAS and its resistance to adaptive attacks. Our code is available here.

## 1. Introduction

Text-to-image (T2I) (Zhang et al., 2025; Wang et al., 2025c) diffusion models have emerged as a major paradigm in generative modeling, achieving breakthrough progress in high-quality image synthesis and being widely adopted in content creation and commercial design. Represented by large-scale pretrained models such as Stable Diffusion (Rombach et al., 2022), T2I diffusion models can generate high-quality and diverse images from user prompts, substantially improving creative efficiency and reshaping content production workflows; moreover, with the advancement of personalization and fine-tuning techniques, these models can be customized to generate images with specific themes or styles (Lim et al., 2025; Wu et al., 2025; Li et al., 2025a). However, training high-performing models typically requires massive data and expensive computational resources (Zheng et al., 2025; Dubiński et al., 2025; Shao et al., 2026), making them valuable intellectual property assets that are also vulnerable to unauthorized copying, redistribution, and misuse (Li et al., 2025d; Lyu et al., 2025; Li et al., 2025b). Therefore, effectively protecting the copyright and ownership of T2I diffusion models has become a critical challenge in generative model security and intellectual property protection (Guo et al., 2024; Chen et al., 2023).

To the best of our knowledge, model ownership verification (MOV) is an important tool for mitigating model-stealing risks, aiming to determine whether a *suspicious model* is stolen from a protected *owner model*. Existing MOV methods are generally categorized into *model fingerprinting* (Gloaguen et al., 2025; Pasquini et al., 2025; Shao et al., 2025a) and *model watermarking* (Shao et al., 2025b; Yang et al., 2025; 2026): Fingerprinting typically embeds verification signals outside the model backbone but often exhibits limited robustness under sophisticated stealing scenarios (Li et al., 2025c; Wang et al., 2025d; Zhu et al., 2025). Therefore, we primarily focus on model watermarking, which trains the owner model to produce verifiable outputs on predefined inputs. Specifically, this paper mainly focuses on the most widely used backdoor-based diffusion watermarking paradigm, which embeds a private trigger during training that activates predefined watermark behavior during generation (Zhao et al., 2023; Wang et al., 2025d), thereby enabling MOV for T2I diffusion models in practice.

Inspired by (Qiao et al., 2025; 2026a), we first revisit existing backdoor-based diffusion watermarking methods. We

[1] Generative AI Lab, College of Computing and Data Science, Nanyang Technological University, Singapore [2] Department of Computer Science and Engineering, Texas A&M University, USA . Correspondence to: Yiming Li <liyiming.tech@gmail.com>, Dacheng Tao <dacheng.tao@gmail.com>.

*Proceedings of the 43${}^{rd}$ International Conference on Machine Learning*, Seoul, South Korea. PMLR 306, 2026. Copyright 2026 by the author(s).

find that these methods often implicitly assume a "faithful" verification process, where a verifier can query a suspicious model and obtain faithful watermark responses to complete MOV. However, in realistic adversarial settings, this assumption frequently breaks down, as an adversary could intentionally or unintentionally damage potential watermark signals. In particular, our experiments demonstrate that both unintentional random perturbations and intentionally crafted adversarial perturbations can substantially corrupt the watermark signal and degrade verification reliability. Meanwhile, for MOV of conventional classifiers, pioneering studies (Bansal et al., 2022; Jiang et al., 2023; Ren et al., 2023) have provided certified robustness guarantees: as long as parameter perturbations are confined within a certified region, the watermark remains stable and non-removable in the worst case. This leads to a key question: *can we design a certified watermark for T2I diffusion models that is both stealthy and equipped with provable robustness guarantees?*

The answer to this question is affirmative, although we cannot simply adapt existing certified watermarking methods from classifiers. This is mainly because diffusion models learn *score estimates* (*i.e.*, gradients of the log-density) over a much broader data region rather than relying on low-dimensional decision boundaries, and certified robust training at this scale is computationally prohibitive. To address this gap, we propose Cert-LAS, the first certified watermarking method for T2I diffusion models for model ownership verification, and show that our method enables reliable MOV under certain conditions (*e.g.*, bounded parameter perturbations). In general, our Cert-LAS operates in two stages. In the first stage, we embed a trigger-free watermark under layer-adaptive smoothing. We begin by allocating layer-wise noise to each UNet layer according to the Layer Fine-tuning Sensitivity (LFS) indicator, concentrating noise on more vulnerable layers to enlarge the certifiable region. We then leverage a private diffusion classifier to induce the generator to produce misclassified watermark signals, while a perceptual consistency regularizer maintains generation quality, evading watermark auditing from both input and output spaces. To make robust optimization computationally tractable for diffusion models, we further adopt an exponential growth schedule that progressively increases the number of noise samples for gradient averaging during training, significantly reducing the computational overhead of robust optimization. In the second stage, we introduce two statistics, *i.e.*, Watermark Robustness (WR) and Reference Probability (RP), to measure the probability of predicting the target class prompt for a suspected watermarked generator and an unwatermarked reference generator under layer-adaptive smoothing, respectively. In particular, we prove a lower bound on their gap when parameter perturbations on the suspected model remain within the certifiable range. As such, by employing a paired-sample

$t$-test, the suspicious model can be verified as derived from the protected model (without authorization) if its WR value is significantly larger than the RP value of a reference generator that is independently trained without watermarking, thereby achieving certified ownership verification.

In summary, the main contributions of this paper are fourfold: **(1)** We revisit existing model ownership verification (MOV) methods for text-to-image (T2I) diffusion models and reveal their limitations in undetectability and robustness; **(2)** We propose Cert-LAS, the first certified watermarking method for T2I diffusion models, which uses a diffusion classifier as an implicit watermark carrier and introduces layer-adaptive randomized smoothing tailored to UNet architectures; **(3)** We theoretically analyze the robustness guarantees of the proposed MOV method for T2I diffusion models and establish the corresponding conditions; **(4)** We conduct extensive experiments to validate the effectiveness of our method and its resistance to potential adaptive attacks.

## 2. Related Work

### 2.1. Copyright Protection in Deep Learning

Copyright protection of deep learning systems mainly proceeds along two complementary lines: *data(set) ownership verification* (DOV) and *model ownership verification* (MOV), depending on the protected asset. DOV (Shao et al., 2025c; Li et al., 2025b) embeds verification signals into the protected data and tests whether a suspect model has been developed based on it. Although DOV can in principle be repurposed for MOV by checking whether a suspect model has been trained on this private dataset, the defender can only watermark the dataset itself and has no control over the training pipeline, making DOV insufficient for reliable model ownership verification against informed adversaries with stronger knowledge of the training pipeline. We therefore focus on MOV, where the defender directly controls the watermark embedding inside the protected model and can thus enforce stronger robustness guarantees.

Existing MOV approaches mainly fall into two categories: model fingerprinting (Gloaguen et al., 2025; Pasquini et al., 2025) and model watermarking (Shao et al., 2025b; Yang et al., 2025), depending on whether predefined secret inputs are needed (see Appendix B). Fingerprinting methods typically embed verification signals outside the model backbone, but they often become less effective under sophisticated stealing scenarios (Gan et al., 2023; Li et al., 2025c; Wang et al., 2025d). Therefore, we primarily focus on model watermarking, which trains the owner model to produce verifiable outputs on specific inputs. In recent years, model watermarking has expanded from image classification (Liu et al., 2023a) to a range of tasks (Yang et al., 2024b; Wang et al., 2025b). With the rapid rise of generative models,

*Table 1.* Prompt and image space suspicious scores (%) for representative backdoor-based watermarking methods.

| Method | $\mathcal{S}_{\text{in}}(p)$ | $\mathcal{S}_{\text{out}}(p)$ |
|---|---|---|
| WatermarkDM | 96.42 | 93.30 |
| SleeperMark | 95.81 | 2.45 |

text-to-image (T2I) diffusion models have also attracted growing attention. In this context, backdoor-based watermarking with synthetic triggers (Zhao et al., 2023; Wang et al., 2025d; Gao et al., 2026) (*i.e.*, rare tokens and atypical patterns) has gradually become the dominant method. However, synthetic triggers are more prone to detection and removal due to their semantic atypicality (Liang et al., 2024a;b), and existing methods lack certified robustness guarantees against adaptive attacks (Wang et al., 2023; Zhu et al., 2024; Wang et al., 2025a). Therefore, it is necessary to develop watermarking approaches that achieve both stealthiness and certified robustness.

### 2.2. Certified Robustness

Certified robustness (Voracek & Hein, 2023; Lyu et al., 2024; Qiao et al., 2026a) guarantees that a model's output remains unchanged under any perturbation within a provable certified region. Randomized smoothing (Cohen et al., 2019; Salman et al., 2019; Qiao et al., 2026b) is a widely used approach to achieve this, injecting random noise and aggregating predictions via majority vote to produce smoothed outputs. This idea has been extended from input space to parameter space (Bansal et al., 2022; Jiang et al., 2023; Ren et al., 2023), enabling certified watermark robustness in MOV: as long as parameter perturbations remain within a certified region, the watermark exhibits provable non-removability. However, these methods are designed for small-scale classifiers with low-dimensional decision boundaries, whereas T2I diffusion models learn score estimates over high-dimensional continuous distributions (Song & Ermon, 2019; Song et al., 2021; Li et al., 2025a), making both fidelity preservation and certified training significantly more challenging. Consequently, certified MOV techniques for conventional classifiers cannot be directly transferred to T2I diffusion models, leaving certified watermarking for T2I diffusion models unexplored.

## 3. Revisiting T2I Backdoor Watermarking

Existing backdoor-based diffusion watermarking methods often (implicitly) assume a "faithful" verification process, in which watermark responses from a queried model are stable enough for reliable verification. However, in real-world settings, the adversary may first conduct watermark auditing to expose embedded patterns and then apply intentional or unintentional perturbations that can corrupt the watermark signal and significantly degrade verification reliability. In this section, we examine whether current backdoor-based dif-

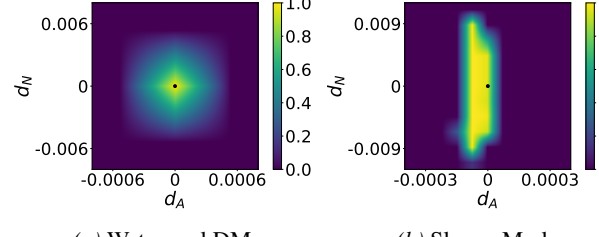

*(a)* WatermarkDM      *(b)* SleeperMark

*Figure 1.* TPR@FPR $= 10^{-6}$ of watermarked models under parameter perturbations. $\mathbf{d}_N$ and $\mathbf{d}_A$ denote the random noise and adversarial directions, respectively, and '●' marks the original watermarked model.

fusion watermarking methods remain effective under these conditions. Before presenting our experiments and results, we briefly review the general workflow of such methods.

**Main Pipeline of (Backdoor-Based) Diffusion MOV Methods.** Consider a diffusion generator $G(\cdot; \boldsymbol{\theta})$ inducing conditional distributions $P_{\boldsymbol{\theta}}(\cdot|p)$ over $\mathcal{X} = [0,1]^{C \times H \times W}$ given prompts $p \in \mathcal{P}$. In the *model watermarking phase*, the owner defines a private trigger $\kappa$ with associated trigger map $T_{\kappa} : \mathcal{P} \to \mathcal{P}$ (*e.g.*, prepending a secret token), and produces a watermarked generator $G(\cdot; \boldsymbol{\theta}_w)$ such that benign behavior is preserved while triggered sampling $P_{\boldsymbol{\theta}_w}(\cdot|T_{\kappa}(p))$ yields images containing a verifiable artifact. The owner also specifies a verification functional $\Psi_{\kappa} : \mathcal{X}^N \to \{0,1\}$ that determines whether $N$ *i.i.d.* triggered samples exhibit the artifact (*e.g.*, via target-image similarity or bit-string decoding). In the *ownership verification phase*, given a suspicious generator $G(\cdot; \boldsymbol{\theta}')$, the owner queries it with triggered prompts $\tilde{p}_i = T_{\kappa}(p_i)$ to obtain $\boldsymbol{x}_i \sim P_{\boldsymbol{\theta}'}(\cdot|\tilde{p}_i)$, and claims ownership if $\Psi_{\kappa}(\boldsymbol{x}_1, \ldots, \boldsymbol{x}_N) = 1$.

### 3.1. Limitations of Trigger Stealthiness under Auditing

**Settings.** We hereby exploit two representative backdoor-based diffusion watermarking methods (*i.e.*, WatermarkDM (Zhao et al., 2023) and SleeperMark (Wang et al., 2025d)) to evaluate the detectability of their synthetic triggers when an adversary audits both prompts and generated images. For **prompt-space auditing**, we measure word-level contextual incongruity. Given a prompt $p = \{w_i\}_{i=1}^n$, we use GPT-2 medium (Radford et al., 2019) to compute for each word the conditional surprisal $m_i(p)$ under the original left context and a baseline surprisal $q_i(p)$ under an unconditional prior. The prompt suspiciousness score is then $\mathcal{S}_{\text{in}}(p) \triangleq \sigma\left(\max_i \frac{m_i(p)}{q_i(p)}\right)$, where $\sigma(\cdot)$ is the logistic sigmoid. When $\mathcal{S}_{\text{in}}(p)$ suggests trigger presence, we conduct **image-space auditing**: we generate images from both the original prompt $p^+ = p$ and a de-triggered variant $p^-$, then compare their within-prompt similarity $\mathcal{M}(p)$ (average pairwise MSE). The image suspiciousness score is $\mathcal{S}_{\text{out}}(p) \triangleq \frac{|\mathcal{M}(p^+) - \mathcal{M}(p^-)|}{\mathcal{M}(p^-)}$. For both metrics, higher scores indicate greater likelihood of trigger presence.

**Results.** As shown in Tab. 1, both methods are detectable but differ in detection space. Specifically, WatermarkDM is detectable in both spaces, with $\mathcal{S}_{\text{in}}(p)$ and $\mathcal{S}_{\text{out}}(p)$ both exceeding 93%. In contrast, SleeperMark evades image-space detection $\mathcal{S}_{\text{out}}(p) = 2.45\%$ due to semantic diversity preservation, but remains highly detectable in prompt space $\mathcal{S}_{\text{in}}(p) > 95\%$. These results reveal an inherent limitation of backdoor-based watermarking: synthetic triggers compromise stealthiness and can be identified through auditing.

### 3.2. Limitations of Watermark Robustness under Adversarial Parameter Perturbations

**Settings.** To examine watermark robustness under perturbations, we visualize $\text{TPR@FPR} = 10^{-6}$ in a two-dimensional parameter subspace. Following previous works (Zhao et al., 2023; Wang et al., 2025d), the verification functionals are method-specific: for WatermarkDM, $\Psi_\kappa$ accepts ownership when the SSIM (Wang et al., 2004) between generated samples and a predefined reference image exceeds a threshold; for SleeperMark, $\Psi_\kappa$ accepts when the bit-decoding accuracy from a pretrained message decoder exceeds a threshold. All thresholds are calibrated via hypothesis testing on unwatermarked models to control FPR at $10^{-6}$. We define the perturbed parameters as $\boldsymbol{\theta}'(\epsilon_N, \epsilon_A) = \boldsymbol{\theta}_w + \epsilon_N \mathbf{d}_N + \epsilon_A \mathbf{d}_A$, where $\mathbf{d}_N$ is a random sign vector and $\mathbf{d}_A$ is the worst-case adversarial direction that maximally degrades verification. These perturbed parameters correspond to the perturbed generator $G(\cdot; \boldsymbol{\theta}'(\epsilon_N, \epsilon_A))$. We report $\text{TPR@FPR} = 10^{-6}$ across the $(\epsilon_N, \epsilon_A)$ grid to illustrate how verification degrades under random versus adversarial perturbations.

**Results.** As shown in Fig. 1, both random and adversarial perturbations degrade watermark verification, although adversarial perturbations are far more effective. For Sleeper-Mark, a random perturbation requires a magnitude of 0.009 to cause a noticeable drop, whereas an adversarial perturbation of only 0.0003 already induces significant degradation. This indicates that while parameter perturbations in general can compromise T2I diffusion watermarks, adversarial perturbations even pose a substantially stronger threat.

## 4. The Proposed Method

As demonstrated in Section 3, existing backdoor-based diffusion watermarking methods are fragile under both watermark auditing and parameter perturbations. To address these issues, we propose Cert-LAS, the first certified, trigger-free MOV method for T2I diffusion models based on layer-adaptive smoothing. Before presenting the method details, we first introduce the threat model and preliminaries.

### 4.1. Threat Model

Following the classical model ownership verification (MOV) setting (Yang et al., 2024b; Shao et al., 2025b; Li et al., 2025c), we consider a model owner (*i.e.*, defender), an adversary, and a trusted verification authority. The model owner embeds a watermark into a text-to-image (T2I) diffusion model before release. After release, users may apply the model to downstream tasks, while adversaries may steal it for unauthorized exploitation and ownership claims (Liang et al., 2022; Guo et al., 2023). Once an infringement is suspected, the verification authority obtains a copy of the suspect model (*e.g.*, its weights) and performs verification using private information held only by the owner; ownership violation is confirmed if the watermark is detected. Unlike prior work that implicitly assumes adversaries only perform benign personalization fine-tuning, we further consider stronger adversaries who may adaptively remove watermarks via post-release parameter modifications, reflecting more realistic adversarial scenarios. An extended discussion is provided in Appendix H.

### 4.2. Preliminaries

In this section, we introduce the theoretical foundations of layer-adaptive noise design. We begin by introducing a metric to quantify layerwise parameter dynamics during fine-tuning, which enables us to formalize layer-adaptive noise. To ensure fair comparison with layer-uniform baselines, we further establish a budget-matching criterion.

**Definition 4.1** (Average $L_2$ Norm). For a diffusion model with $L$ layers of dimensions $\{d_l\}_{l=1}^L$, let $\boldsymbol{\theta}^l \in \mathbb{R}^{d_l}$ represent the parameters of layer $l$. The average $L_2$ norm measures the magnitude of parameter change for layer $l$ between step $t_1$ and step $t_2$: $\bar{\delta}^l(t_1, t_2) \triangleq \frac{\left\| \boldsymbol{\theta}^l(t_2) - \boldsymbol{\theta}^l(t_1) \right\|_2}{\sqrt{d_l}}$, where $\boldsymbol{\theta}^l(t)$ denotes the parameter vector of layer $l$ at step $t$ and $\|\cdot\|_2$ represents the $L_2$ norm.

Next, we formalize layer-adaptive noise as follows.

**Definition 4.2** (Layer-adaptive Noise). For a scaling factor $k$, let $\boldsymbol{\epsilon}_k = (\boldsymbol{\epsilon}_k^1, \ldots, \boldsymbol{\epsilon}_k^L) \sim \mathcal{E}_k$ denote a noise sample with $\boldsymbol{\epsilon}_k^l \in \mathbb{R}^{d_l}$. We call $\mathcal{E}_k$ a *layer-adaptive* noise distribution if $\boldsymbol{\epsilon}_k$ is non-uniform with respect to the layer-block decomposition, *i.e.*, there exist layers $l \neq j$ such that the marginal distributions of $\boldsymbol{\epsilon}_k^l$ and $\boldsymbol{\epsilon}_k^j$ have different scales under the Euclidean norm.

*Remark* 4.3 (Multivariate Gaussian Specialization). In this paper, we instantiate Definition 4.2 with a multivariate Gaussian distribution, since it is analytically tractable for certification and often serves as a common approximation to aggregated stochastic effects. Let $\boldsymbol{\sigma}_k = (k\sigma_1, \ldots, k\sigma_L)$ denote the scaled layer-wise noise levels. We take $\boldsymbol{\epsilon}_k \sim \mathcal{N}(\mathbf{0}, \boldsymbol{\Sigma}_k)$ with block diagonal covariance $\boldsymbol{\Sigma}_k = \text{diag}(\sigma_{k,1}^2 \mathbf{I}_{d_1}, \ldots, \sigma_{k,L}^2 \mathbf{I}_{d_L})$, where $\sigma_{k,l} = k\sigma_l$.

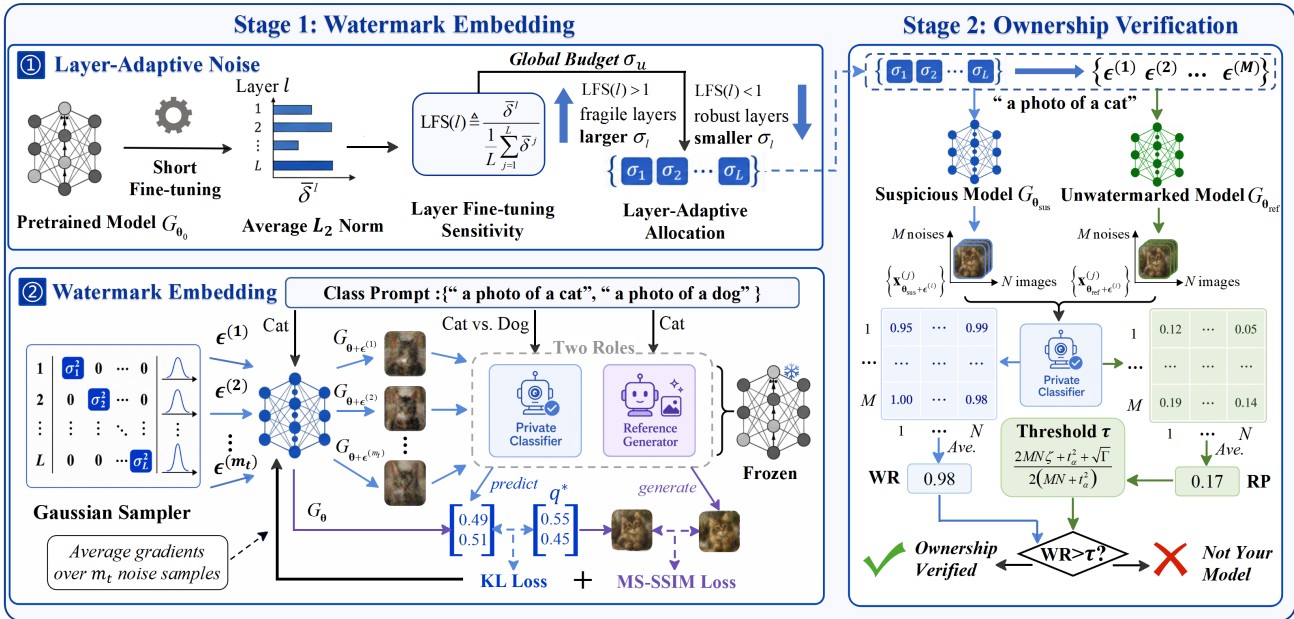

*Figure 2.* The main pipeline of Cert-LAS consists of two stages. In the first stage, we conduct a short fine-tuning on the pretrained model $G_{\boldsymbol{\theta}_0}$ to derive the Layer Fine-tuning Sensitivity LFS($l$), which adaptive assigns layer-wise noise levels $\{\sigma_1, \ldots, \sigma_L\}$ under a global noise budget $\sigma_u$, allocating larger noise to fragile layers. Guided by this allocation, we sample layer-adaptive noise $\boldsymbol{\epsilon}$ and inject it into the generator, and leverage a frozen model as a private diffusion classifier, training the generator to induce misclassification toward a target posterior $q^*$ while preserving fidelity. In the second stage, we reuse the layer-adaptive noise levels to compute the Watermark Robustness (WR) of a suspicious model and the Reference Probability (RP) of an unwatermarked reference; the suspicious model is verified as derived from the protected one if WR significantly exceeds RP, equivalently when WR surpasses the closed-form threshold in Eq. (5).

To enable fair comparison with layer-uniform baselines, we define a *budget equivalence condition* matching the total noise budget across allocation strategies.

**Definition 4.4** (Budget Equivalence Condition). Let $\boldsymbol{\sigma} = (\sigma_1, \ldots, \sigma_L)$ specify a layer-adaptive noise as in Definition 4.2, and let $\sigma_u$ denote the noise level of a layer-uniform noise. We say the two noises are *budget-matched* if they induce equivalent Mahalanobis distances for all parameter changes, which requires: $\sigma_u^2 = \frac{\sum_{l=1}^{L} d_l \sigma_l^2}{\sum_{l=1}^{L} d_l}$.

Intuitively, Definition 4.4 ensures that layer-uniform and layer-adaptive schemes use equivalent total noise budget, so any performance difference can be attributed solely to the noise allocation strategy.

### 4.3. Overview of the Proposed Method

In general, our Cert-LAS consists of two main stages, as illustrated in Fig. 2: **(1)** watermark embedding and **(2)** ownership verification. In the first stage, we embed a trigger-free watermark by training the diffusion generator under layer-adaptive smoothing, where noise is allocated across layers according to a Layer Fine-tuning Sensitivity (LFS) indicator. To handle the computational burden of robust optimization, we employ an exponential growth schedule that progressively increases the number of noise samples. In the second stage, we estimate the Watermark

Robustness (WR) of the suspicious generator and the Reference Probability (RP) of an unwatermarked reference, then perform ownership verification via a paired-sample statistical test. The suspicious model is verified as watermarked if WR is significantly larger than RP at a certain significance level. The technical details are as follows.

### 4.4. Watermark Embedding of Cert-LAS

In this stage, we present the watermark embedding procedure. Since the UNet architecture exhibits layer-dependent sensitivity to fine-tuning, we derive a layer-adaptive noise allocation that concentrates noise on vulnerable layers, thereby enlarging the certified radius. To embed watermarks under such smoothing without triggers, we leverage diffusion models as private classifiers, enabling trigger-free verification while remaining stealthy against auditing. Optimizing this objective under layer-adaptive smoothing would naively require averaging gradients over many noise samples. We therefore adopt an exponential growth schedule that gradually increases the gradient averaging frequency, substantially reducing computational overhead.

**Layer-Adaptive Noise.** To address the architectural differences between discriminative classifiers and generative diffusion models, we study how different UNet layers respond to downstream fine-tuning (see Appendix C). We observe that the relative magnitudes of layerwise updates

remain consistent across diverse datasets, which may stem from UNet's characteristic *compress-then-generate* architecture. Based on this observation, we introduce a layerwise sensitivity indicator to guide noise allocation.

**Definition 4.5** (Layer Fine-tuning Sensitivity)**.** The fine-tuning sensitivity of layer $l$ is defined as the ratio of its update magnitude to the average update magnitude across all layers: $\mathrm{LFS}(l) \triangleq \frac{\bar{\delta}^l(0,T)}{\frac{1}{L}\sum_{j=1}^{L} \bar{\delta}^j(0,T)}$, where $\bar{\delta}^l(0,T)$ denotes the average $L_2$ norm of parameter changes in layer $l$ from initialization to step $T$.

Intuitively, $\mathrm{LFS}(l)$ measures the *normalized* update magnitude of layer $l$ under downstream fine-tuning. Under a fixed global noise budget, layers with $\mathrm{LFS}(l) > 1$ receive a larger noise level and vice versa, leading to the following layer-adaptive allocation.

**Proposition 4.6** (Layer-Adaptive Noise Allocation)**.** *Consider a layer-adaptive Gaussian noise with noise levels proportional to fine-tuning sensitivity, i.e., $\sigma_l \propto \mathrm{LFS}(l)$. Under the global noise budget $\sigma_\mathrm{u}$, the allocation*

$$\sigma_l = \sigma_\mathrm{u} \cdot \mathrm{LFS}(l) \cdot \sqrt{\frac{\sum_{j=1}^{L} d_j}{\sum_{j=1}^{L} d_j \, \mathrm{LFS}(j)^2}}$$

*is non-uniform whenever $\mathrm{LFS}(l)$ varies across layers, and satisfies the equivalence condition in Definition 4.4.*

In particular, this allocation assigns a higher noise level to vulnerable layers than layer-uniform smoothing while keeping the total noise budget unchanged.

**Trigger-Free Watermark.** Inspired by recent findings that diffusion models can function as generative classifiers (Li et al., 2023; Chen et al., 2024; Clark & Jaini, 2023), we propose to embed watermarks by training the generator to induce misclassification under a private diffusion classifier, enabling implicit watermark verification without triggers (*i.e.*, using only class prompts). Since modifying the generation distribution may degrade generation fidelity, we further encourage the watermarked generator to retain perceptual fidelity to the original. To formalize this mechanism, we first define the diffusion classifier as an energy-based model.

**Definition 4.7** (Diffusion Classifier as Energy-Based Model)**.** Let $x \in \mathcal{X}$ be an input image and $y \in \mathcal{Y}$ a class label. A diffusion model parameterized by $\boldsymbol{\theta}$ with noise predictor $\varepsilon_{\boldsymbol{\theta}}$ induces an energy function $E_{\boldsymbol{\theta}}(x, y) \triangleq \mathbb{E}_{t,\boldsymbol{\eta}}\left[\|\boldsymbol{\eta} - \varepsilon_{\boldsymbol{\theta}}(x_t, y)\|_2^2\right]$, where $x_t$ denotes the noise perturbed input at diffusion timestep $t$. The corresponding Gibbs posterior is given by

$$p_{\boldsymbol{\theta}}(y \mid x) = \frac{\exp(-E_{\boldsymbol{\theta}}(x, y))}{\sum_{\hat{y}\in\mathcal{Y}} \exp(-E_{\boldsymbol{\theta}}(x, \hat{y}))}. \quad (1)$$

As formalized in Definition 4.7, a diffusion model induces a well-defined posterior $p_{\boldsymbol{\phi}}(y \mid x)$ via its energy function, which we leverage for watermark embedding. Specifically, given a frozen private diffusion classifier parameterized by $\boldsymbol{\phi}$ and a binary prompt set $\mathcal{Y} = \{y, y'\}$, we train the generator $G(\cdot; \boldsymbol{\theta})$ such that samples $x \sim P_{\boldsymbol{\theta}}(\cdot \mid y)$ induce $p_{\boldsymbol{\phi}}(\cdot \mid x)$ matching a target distribution $q^*$. The watermark objective minimizes the Kullback–Leibler divergence:

$$\mathcal{L}_{\mathrm{KL}}(\boldsymbol{\theta}) = \mathbb{E}_{x \sim P_{\boldsymbol{\theta}}(\cdot|y)}\left[D_{\mathrm{KL}}\left(q^* \,\|\, p_{\boldsymbol{\phi}}(\cdot \mid x)\right)\right]. \quad (2)$$

Since optimizing $\mathcal{L}_{\mathrm{KL}}$ alone may alter the generation distribution and degrade generation quality, we further regularize $G(\cdot; \boldsymbol{\theta})$ against $G(\cdot; \boldsymbol{\phi})$, reusing the frozen diffusion model as a reference generator. For each class prompt $y \in \mathcal{Y}$, we draw $x_{\boldsymbol{\theta}} \sim P_{\boldsymbol{\theta}}(\cdot \mid y)$ and $x_{\boldsymbol{\phi}} \sim P_{\boldsymbol{\phi}}(\cdot \mid y)$, and minimize their MS-SSIM discrepancy, which measures luminance, contrast, and structure across multiple scales:

$$\mathcal{L}_{\mathrm{ssim}}(\boldsymbol{\theta}) = \mathbb{E}_{y\sim\mathcal{Y}}\mathbb{E}_{x_{\boldsymbol{\theta}},x_{\boldsymbol{\phi}}}\left[1 - \mathrm{MS\text{-}SSIM}(x_{\boldsymbol{\theta}}, x_{\boldsymbol{\phi}})\right]. \quad (3)$$

With the perceptual regularization, the training objective at step $t$ is: $\boldsymbol{\theta}_t^* \in \arg\min_{\boldsymbol{\theta}} \mathcal{L}_{\mathrm{KL}}(\boldsymbol{\theta}) + \mathcal{L}_{\mathrm{ssim}}(\boldsymbol{\theta})$.

**Exponential Growth Schedule.** However, layer-adaptive smoothing requires averaging gradients over many noise samples at each update, making it computationally expensive. Fortunately, pretrained T2I diffusion models exhibit strong generative priors whose subspace lies in flat loss basins (Karras et al., 2024; Ma et al., 2025; Mao et al., 2025), meaning that noise samples produce nearly identical gradients in the early training phase. Intensive smoothing therefore offers little benefit until the model begins to induce classifier misclassification. Accordingly, we adopt an exponential schedule that starts with minimal smoothing and progressively increases both the number of noise samples $m_t$ and the regularization strength $\omega_t$: $m_t = \min\left(m_{\max}, \lfloor 2^{t/T_\mathrm{g}} \rfloor\right)$, $\omega_t = \omega_0 \cdot 2^{t/T_\mathrm{g}}$, where $\omega_0$ is the initial regularization weight and $T_\mathrm{g}$ controls the doubling period.

### 4.5. Ownership Verification of Cert-LAS

In this stage, inspired by certified dataset ownership verification (Qiao et al., 2025; 2026a), we introduce two statistics under layer-adaptive smoothing: Watermark Robustness (WR) and Reference Probability (RP). Specifically, WR measures the empirical probability that images generated by the suspected model are classified as the target class prompt by the diffusion classifier, while RP measures the same probability for an unwatermarked reference generator serving as a baseline. Since the embedded watermark persists under parameter perturbations, a watermarked generator will exhibit WR significantly higher than RP. To formalize this comparison, we employ a paired-sample $t$-test (Student, 1908) and derive a closed-form threshold on WR. Ownership of the

suspected model is then verified whenever WR exceeds this threshold. The proof is deferred to Appendix A.

**Computing WR and RP under Layer-Adaptive Smoothing.** Given a target prompt $\tilde{y} \in \mathcal{Y}$, let $q_\phi(\boldsymbol{x}) \triangleq \arg\max_{c \in \mathcal{Y}} p_\phi(c \mid \boldsymbol{x})$ denote the class prediction of the diffusion classifier. For a suspected generator $G(\cdot; \boldsymbol{\theta}_{\text{sus}})$ and an unwatermarked reference generator $G(\cdot; \boldsymbol{\theta}_{\text{ref}})$, we generate samples $\boldsymbol{x}_{\boldsymbol{\theta}+\boldsymbol{\epsilon}_k^{(i)}}^{(j)} \sim P_{\boldsymbol{\theta}+\boldsymbol{\epsilon}_k^{(i)}}(\cdot \mid \tilde{y})$ under layer-adaptive noise $\boldsymbol{\epsilon}_k$. We define the WR of $G_{\boldsymbol{\theta}_{\text{sus}}}$ and the RP of $G_{\boldsymbol{\theta}_{\text{ref}}}$ as:

$$\text{WR}(G_{\boldsymbol{\theta}_{\text{sus}}}, \mathcal{E}_k) \triangleq \frac{1}{N}\sum_{j=1}^{N} \frac{1}{M}\sum_{i=1}^{M} \mathbb{I}\left\{ q_\phi\left( \boldsymbol{x}_{\boldsymbol{\theta}_{\text{sus}}+\boldsymbol{\epsilon}_k^{(i)}}^{(j)} \right) = \tilde{y} \right\},$$

$$\text{RP}(G_{\boldsymbol{\theta}_{\text{ref}}}, \mathcal{E}_k) \triangleq \frac{1}{N}\sum_{j=1}^{N} \frac{1}{M}\sum_{i=1}^{M} \mathbb{I}\left\{ q_\phi\left( \boldsymbol{x}_{\boldsymbol{\theta}_{\text{ref}}+\boldsymbol{\epsilon}_k^{(i)}}^{(j)} \right) = \tilde{y} \right\},$$

(4)

where $\mathbb{I}\{\cdot\}$ denotes the indicator function. In practice, both quantities are estimated with $M$ i.i.d. draws $\boldsymbol{\epsilon}_k^{(i)} \overset{\text{i.i.d.}}{\sim} \mathcal{E}_k$ over $N$ verification samples. After obtaining WR and RP, model ownership is then verified through the paired-sample $t$-test.

### 4.6. Theoretical Analysis of Cert-LAS

In this section, we establish the theoretical analysis of ownership verification of Cert-LAS proposed in Section 4.5. The detailed proof is provided in Appendix A.

**Theorem 4.8** (Closed-Form Threshold for Ownership Verification). *Consider testing the null hypothesis $H_0 : \text{WR} = \text{RP}$ against the alternative $H_1 : \text{WR} > \text{RP}$. Given an upper bound $\text{RP} \leq \zeta$, at significance level $\alpha$, the suspicious model $G(\cdot; \boldsymbol{\theta}_{\text{sus}})$ is verified as watermarked whenever*

$$\text{WR} > \frac{2MN\zeta + t_\alpha^2 + \sqrt{\Gamma}}{2(MN + t_\alpha^2)},$$

(5)

*where $\Gamma = (2MN\zeta + t_\alpha^2)^2 - 4(MN + t_\alpha^2)(MN\zeta^2 - t_\alpha^2\zeta + t_\alpha^2\zeta^2)$, and $t_\alpha$ denotes the $(1-\alpha)$-quantile of the $t$-distribution with $(N-1)$ degrees of freedom.*

**Theorem 4.9** (Certified Radius under Layer-Adaptive Gaussian Smoothing). *Consider Gaussian smoothing noise $\mathcal{E}_k = \mathcal{N}(\mathbf{0}, \boldsymbol{\Sigma}_k)$ and the normalized radius $\bar{R}$ as in Definition A.2. For probability thresholds $a \leq s_1 \leq \cdots \leq s_m \leq b$, let $P_{s_j}(\boldsymbol{\theta}) \leq \mathbb{P}_{\boldsymbol{\epsilon}_k \sim \mathcal{E}_k}\left[ \frac{1}{N}\sum_{i=1}^{N} \mathbb{I}\left\{ q_\phi\left( \boldsymbol{x}_{\boldsymbol{\theta}+\boldsymbol{\epsilon}_k}^{(i)} \right) = \tilde{y} \right\} \geq s_j \right]$. For any perturbation $\boldsymbol{\delta} \in \mathcal{B}_{\boldsymbol{\sigma}_k}(\boldsymbol{\theta}; R_k)$, diffusion model ownership verification with layer-adaptive noise is guaranteed if $\bar{R} \leq R^*$, which is obtained by solving*

$$a + (s_1 - a)\Phi\left( \Phi^{-1}(P_{s_1}(\boldsymbol{\theta})) - \frac{\bar{R}}{k} \right)$$
$$+ \sum_{j=2}^{m}(s_j - s_{j-1})\Phi\left( \Phi^{-1}(P_{s_j}(\boldsymbol{\theta})) - \frac{\bar{R}}{k} \right) > \tau_{\alpha,\zeta},$$

(6)

*where $\Phi$ is the standard Gaussian CDF, $\alpha$ is the significance level and $\tau_{\alpha,\zeta} = \frac{2MN\zeta + t_\alpha^2 + \sqrt{\Gamma}}{2(MN + t_\alpha^2)}$ is the verification threshold in Theorem 4.8.*

In general, Theorem 4.8 first establishes a closed-form verification threshold $\tau_{\alpha,\zeta}$, which then serves as the decision boundary in Theorem 4.9 to derive the certified radius $R^*$. Since a higher WR relaxes the constraint in Eq. (6), stronger watermark embedding directly yields a larger certified radius $R^*$, implying that stronger watermark robustness leads to greater tolerance against parameter perturbations.

## 5. Experiments

### 5.1. Experiment Setup

**Diffusion Models.** Following prior work (Fernandez et al., 2023; Zhao et al., 2023; Wang et al., 2025d), we implement Cert-LAS on Stable Diffusion v1.4 (SD v1.4) (Rombach et al., 2022), the standard benchmark in diffusion watermarking research, to enable fair and direct comparison with existing methods. Our method employs two instances: one serves as the protected generative model that embeds the watermark, while the other remains frozen and serves as both the private diffusion classifier and the reference generator.

**Watermark Removal Attacks.** We evaluate the robustness of Cert-LAS against both unintentional and intentional removal attacks. For unintentional removal, we consider standard fine-tuning on diverse datasets as well as advanced fine-tuning methods including LoRA (Hu et al., 2022), Dream-Booth (Ruiz et al., 2023), and Custom Diffusion (Kumari et al., 2023). For intentional removal, we consider adaptive attacks including $\ell_2$-bounded PGD attacks in parameter space and parameter perturbations along both adversarial and random directions. Details are provided in Appendix D.

**Evaluation Metrics.** We evaluate Cert-LAS on model fidelity and watermark effectiveness. For fidelity, following prior works (Song et al., 2024; Kang et al., 2024), we report FID (Parmar et al., 2022), CLIP score (Radford et al., 2021), and DreamSim (Fu et al., 2023) on 50,000 images generated from COCO2014 (Lin et al., 2014) validation captions, where DreamSim compares images generated from the watermarked and pretrained models. For watermark effectiveness, we adopt metrics tailored to two method paradigms. For empirical methods, we report the confidence-based T@$10^{-6}$F for fair comparison with empirical baselines, together with suspiciousness scores $\mathcal{S}_{\text{in}}(p)$ and $\mathcal{S}_{\text{out}}(p)$ under watermark auditing to assess stealthiness. For certified methods, we report the label-based verification success rate (VSR) and the certified radius $\bar{R}$, which quantify certified verifiability and provable robustness. Further details are provided in Appendix D.

*Table 2.* Performance comparison between Cert-LAS and baseline methods. Cert-LAS (w/o smooth) and Cert-LAS (w/ smooth) denote our method without and with inference-time layer-adaptive smoothing, respectively. Top results within each category are in bold.

| Category | | Method | Model Fidelity | | | Watermark Effectiveness | | |
|---|---|---|---|---|---|---|---|---|
| | | | FID ↓ | CLIP ↑ | DreamSim ↓ | T@$10^{-6}$F ↑ | $\mathcal{S}_{in}(p)$ ↓ | $\mathcal{S}_{out}(p)$ ↓ |
| Baseline | | None | 25.23 | 31.22 | *N/A* | *N/A* | *N/A* | *N/A* |
| Empirical Methods | | WatermarkDM | 26.27 | 30.86 | 0.164 | 0.879 | 0.964 | 0.933 |
| | | SleeperMark | **25.85** | **31.21** | 0.120 | 0.998 | 0.958 | 0.025 |
| | | **Cert-LAS (w/o smooth)** | 25.91 | 31.13 | **0.111** | **1.000** | **0.125** | **0.016** |
| | $\sigma_k$ | **Method** | FID ↓ | CLIP ↑ | DreamSim ↓ | T@$10^{-6}$F ↑ | VSR ↑ | $\bar{R}$ ↑ |
| Certified Methods | $\sigma_{1.0}$ | Vanilla | 27.48 | 31.05 | 0.123 | 0.944 | 0.812 | 0.81 |
| | | **Cert-LAS (w/ smooth)** | **25.91** | **31.13** | **0.111** | **1.000** | **1.000** | **1.48** |
| | $\sigma_{1.2}$ | Vanilla | **26.33** | **31.13** | **0.101** | 0.718 | 0.647 | 0.45 |
| | | **Cert-LAS (w/ smooth)** | 26.54 | 31.10 | 0.122 | **1.000** | **0.938** | **1.68** |
| | $\sigma_{1.4}$ | Vanilla | **25.82** | **31.14** | **0.087** | 0.281 | 0.385 | 0.00 |
| | | **Cert-LAS (w/ smooth)** | 26.96 | 31.08 | 0.125 | **1.000** | **0.847** | **1.428** |

*Table 3.* T@$10^{-6}$F of different watermarking methods after fine-tuning watermarked SD v1.4 via LoRA.

| LoRA Rank | 20 | | | 80 | | | 320 | | | 640 | | |
|---|---|---|---|---|---|---|---|---|---|---|---|---|
| Fine-tuning Steps | 20 | 200 | 2000 | 20 | 200 | 2000 | 20 | 200 | 2000 | 20 | 200 | 2000 |
| WatermarkDM | 0.893 | 0.687 | 0.000 | 0.885 | 0.714 | 0.000 | 0.865 | 0.588 | 0.000 | 0.830 | 0.451 | 0.000 |
| SleeperMark | 0.998 | 0.998 | 0.996 | 0.998 | 0.997 | 0.993 | 0.998 | 0.997 | 0.990 | 0.998 | 0.996 | 0.988 |
| **Cert-LAS (w/o smooth)** | 1.000 | 1.000 | 1.000 | 1.000 | 1.000 | 1.000 | 1.000 | 1.000 | 1.000 | 1.000 | 1.000 | 0.996 |
| **Cert-LAS (w/ smooth)** | **1.000** | **1.000** | **1.000** | **1.000** | **1.000** | **1.000** | **1.000** | **1.000** | **1.000** | **1.000** | **1.000** | **1.000** |

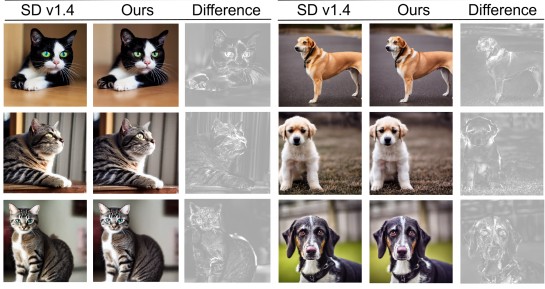

*Figure 3.* Qualitative comparison between images generated by Cert-LAS and the source model under class prompts.

**Baselines.** We compare Cert-LAS with two categories of baselines. For empirical methods, we consider WatermarkDM (Zhao et al., 2023) and SleeperMark (Wang et al., 2025d), evaluated without inference-time smoothing; their original metrics are converted into T@$10^{-6}$F (see Appendix D). For certified methods, we compare against a vanilla layer-uniform smoothing baseline that applies smoothing during verification, with certified radius computed analytically as (Cohen et al., 2019; Jiang et al., 2023).

**Implementation Details.** In this paper, we use cat and dog as the class prompts for diffusion classification following (Li et al., 2023; Chen et al., 2024). The uniform noise level is set to $\sigma_u = 0.01$, with $\sigma_1$ calibrated to match the same noise budget. Further details are provided in Appendix D.

## 5.2. Main Results

As shown in Tab. 2, for watermarked methods, Cert-LAS achieves the best watermark effectiveness with T@$10^{-6}$F of

1.000 and stealthiness scores below 0.125, less than 15% of WatermarkDM. Meanwhile, it maintains competitive fidelity with the smallest DreamSim of 0.111, and images generated under class prompts are perceptually indistinguishable from the source model as shown in Fig. 3. For certified methods, Cert-LAS consistently outperforms Vanilla across all noise levels. As $\sigma_k$ increases, Cert-LAS exhibits mildly degraded fidelity but maintains strong verification, achieving VSR of 0.847 at $k = 1.4$. In contrast, Vanilla exhibits an opposite trend where fidelity improves as verification collapses, indicating it fails to learn the watermark under layer-uniform noise. These results confirm that our method demonstrates strong effectiveness while preserving fidelity.

## 5.3. Resistance to Watermark Removal Attacks

We hereby evaluate the robustness of Cert-LAS against both unintentional and intentional removal attempts.

**Unintentional Attack.** As shown in Tab. 4-6, Cert-LAS demonstrates exceptional robustness against benign downstream fine-tuning across all scenarios. Even under the most challenging regime of standard fine-tuning with full-parameter updates for 2000 steps, Cert-LAS maintains T@$10^{-6}$F above 0.991, whereas WatermarkDM fails entirely after merely 400 steps on CelebA-HQ. Notably, Cert-LAS still maintains T@$10^{-6}$F above 0.987 without smoothing, indicating that our robust training inherently introduces resilience into the watermarks, and inference-time smoothing further strengthens it through majority voting. Moreover, our watermark embedding does not impair downstream adaptation capability (see Fig. 10 in Appendix J).

*Table 4.* T@$10^{-6}$F of different watermarking methods after fine-tuning watermarked SD v1.4 on (a) Cartoon, (b) CelebA-HQ, (c) Landscape, and (d) ArtBench datasets.

| (a) Cartoon | | | | | |
| --- | --- | --- | --- | --- | --- |
| Fine-tuning Steps | 400 | 800 | 1200 | 1600 | 2000 |
| WatermarkDM | 0.716 | 0.432 | 0.179 | 0.000 | 0.000 |
| SleeperMark | 0.998 | 0.998 | 0.998 | 0.997 | 0.993 |
| **Cert-LAS (w/o smooth)** | 1.000 | 1.000 | 0.999 | 0.997 | 0.992 |
| **Cert-LAS (w/ smooth)** | **1.000** | **1.000** | **1.000** | **0.999** | **0.996** |
| (b) CelebA-HQ | | | | | |
| WatermarkDM | 0.137 | 0.000 | 0.000 | 0.000 | 0.000 |
| SleeperMark | 0.998 | 0.996 | 0.995 | 0.989 | 0.984 |
| **Cert-LAS (w/o smooth)** | 1.000 | 1.000 | 1.000 | 1.000 | 0.998 |
| **Cert-LAS (w/ smooth)** | **1.000** | **1.000** | **1.000** | **1.000** | **1.000** |
| (c) Landscape | | | | | |
| WatermarkDM | 0.667 | 0.589 | 0.344 | 0.155 | 0.000 |
| SleeperMark | 0.998 | 0.998 | 0.998 | 0.997 | 0.995 |
| **Cert-LAS (w/o smooth)** | 1.000 | 1.000 | 1.000 | 1.000 | 1.000 |
| **Cert-LAS (w/ smooth)** | **1.000** | **1.000** | **1.000** | **1.000** | **1.000** |
| (d) ArtBench | | | | | |
| WatermarkDM | 0.105 | 0.031 | 0.000 | 0.000 | 0.000 |
| SleeperMark | 0.998 | 0.997 | 0.997 | 0.990 | 0.988 |
| **Cert-LAS (w/o smooth)** | 1.000 | 1.000 | 1.000 | 0.995 | 0.987 |
| **Cert-LAS (w/ smooth)** | **1.000** | **1.000** | **1.000** | **0.998** | **0.991** |

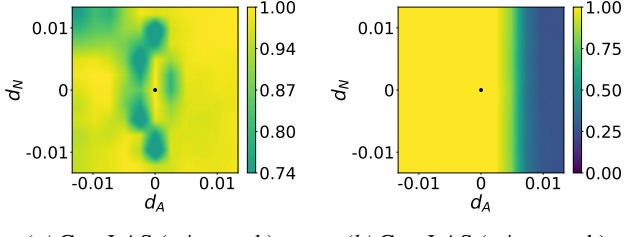

*(a)* Cert-LAS (w/ smooth)  *(b)* Cert-LAS (w/o smooth)

*Figure 4.* T@$10^{-6}$F of watermarked models under parameter perturbations. $\mathbf{d}_N$ and $\mathbf{d}_A$ denote the random noise and adversarial directions, respectively, and '•' marks the original model.

**Intentional Attack.** As shown in Fig. 4 and Tab. 7, baseline methods and Cert-LAS without smoothing exhibit rapid degradation under parameter perturbations. In contrast, Cert-LAS with smoothing eliminates the tenfold asymmetry between random and adversarial perturbations observed in Section 3.2, maintaining T@$10^{-6}$F above 0.965 even at L2 norm budget 0.8 where all baselines fail entirely, demonstrating the robustness against intentional attacks.

**5.4. Ablation Study and Generalization Analysis**

**Ablation Study.** We conduct comprehensive ablation studies along two complementary dimensions: internal hyperparameters (e.g., $\omega_0$ and $q^*$) and external verification components (e.g., the reference generator and private classifier). Notably, the Exponential Growth Schedule reduces training time by 76.6% without sacrificing performance, and the verification signal is uniquely bound to the watermarking-stage private classifier, where replacing it at verification time causes verification to fail across all tested architectures,

*Table 5.* T@$10^{-6}$F of different watermarking methods after fine-tuning watermarked SD v1.4 via DreamBooth.

| Fine-tuning Steps | 200 | 400 | 600 | 800 | 1000 |
| --- | --- | --- | --- | --- | --- |
| WatermarkDM | 0.823 | 0.702 | 0.255 | 0.328 | 0.209 |
| SleeperMark | 0.998 | 0.997 | 0.993 | 0.987 | 0.988 |
| **Cert-LAS (w/o smooth)** | 1.000 | 1.000 | 1.000 | 1.000 | 0.999 |
| **Cert-LAS (w/ smooth)** | **1.000** | **1.000** | **1.000** | **1.000** | **1.000** |

*Table 6.* T@$10^{-6}$F of different watermarking methods after fine-tuning watermarked SD v1.4 via Custom Diffusion.

| Fine-tuning Steps | 100 | 200 | 300 | 400 | 500 |
| --- | --- | --- | --- | --- | --- |
| WatermarkDM | 0.719 | 0.510 | 0.252 | 0.058 | 0.000 |
| SleeperMark | 0.998 | 0.998 | 0.995 | 0.992 | 0.991 |
| **Cert-LAS (w/o smooth)** | 1.000 | 1.000 | 1.000 | 1.000 | 1.000 |
| **Cert-LAS (w/ smooth)** | **1.000** | **1.000** | **1.000** | **1.000** | **1.000** |

*Table 7.* T@$10^{-6}$F of different watermarking methods under $\ell_2$-bounded parameter perturbation on watermarked SD v1.4.

| L2 Norm Budget | 0.2 | 0.4 | 0.6 | 0.8 |
| --- | --- | --- | --- | --- |
| WatermarkDM | 0.781 | 0.000 | 0.000 | 0.000 |
| SleeperMark | 0.999 | 0.861 | 0.060 | 0.000 |
| **Cert-LAS (w/o smooth)** | 0.000 | 0.000 | 0.000 | 0.000 |
| **Cert-LAS (w/ smooth)** | **1.000** | **1.000** | **0.984** | **0.965** |

indicating that the watermark cannot be reproduced by any other classifier. Detailed results are provided in Appendix F.

**Generalization Analysis.** We further examine the generality of Cert-LAS along two axes: model architecture and classification task. For the former, the layerwise update consistency underlying LFS holds across mainstream architectures; for the latter, the broad task level generality, combined with the classifier specificity above, enables non-interference among multiple owners (Appendix I). Detailed results are provided in Appendix G.1.

## 6. Conclusion

In this paper, we revealed the vulnerabilities of existing model ownership verification (MOV) methods for text-to-image diffusion models under watermark auditing and parameter perturbations. To address these issues, we proposed Cert-LAS, the first certified watermarking method based on layer-adaptive smoothing. Cert-LAS leverages diffusion classifiers for trigger-free embedding to evade auditing, and introduces Layer Fine-tuning Sensitivity (LFS) guided noise allocation to enlarge the certifiable region. Our theoretical analysis establishes a tight lower bound between Watermark Robustness (WR) and Reference Probability (RP) under bounded parameter perturbations, enabling certified verification via paired-sample $t$-test. Extensive experiments validate that Cert-LAS maintains high fidelity while achieving superior robustness, paving the way for trustworthy sharing of diffusion models with certified guarantees.

# Acknowledgements

This research is supported by the National Research Foundation, Singapore, and Cyber Security Agency of Singapore under its National Cybersecurity R&D Programme and CyberSG R&D Cyber Research Programme Office. Any opinions, findings and conclusions or recommendations expressed in these materials are those of the author(s) and do not reflect the views of National Research Foundation, Singapore, Cyber Security Agency of Singapore as well as CyberSG R&D Programme Office, Singapore.

# Impact Statement

This paper focuses on protecting the intellectual property of text-to-image diffusion models through certified model ownership verification. We demonstrate that existing backdoor-based watermarking methods are fragile under watermark auditing and subsequent parameter perturbations, and propose Cert-LAS to enable reliable ownership verification even in the presence of bounded, malicious removal attacks. The method is purely defensive, intended solely for legitimate verification by authorized model owners in conjunction with trusted verification authorities when necessary, with strict protection of private verification artifacts. Given that training large-scale diffusion models demands substantial computational and data costs, robust ownership verification helps underpin enforceable licensing and accountability mechanisms, ensuring that creators receive appropriate recognition and compensation while discouraging unauthorized exploitation. Overall, by providing provable robustness guarantees against watermark removal, this work contributes positively to trustworthy intellectual property governance in the generative AI ecosystem and promotes the sustainable development of accountable generative systems.

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

# A. Theoretical Proofs

## A.1. Proof of Theorem 4.8

**Theorem A.1** (Closed-Form Threshold for Ownership Verification). *Consider testing the null hypothesis* $H_0 : \mathrm{WR} = \mathrm{RP}$ *against the alternative* $H_1 : \mathrm{WR} > \mathrm{RP}$. *Given an upper bound* $\mathrm{RP} \leq \zeta$, *at significance level* $\alpha$, *the suspicious model* $G(\cdot; \boldsymbol{\theta}_{\mathrm{sus}})$ *is verified as watermarked whenever*

$$\mathrm{WR} > \frac{2MN\zeta + t_\alpha^2 + \sqrt{\Gamma}}{2(MN + t_\alpha^2)}, \quad (A.1)$$

*where* $\Gamma = (2MN\zeta + t_\alpha^2)^2 - 4(MN + t_\alpha^2)(MN\zeta^2 - t_\alpha^2\zeta + t_\alpha^2\zeta^2)$, *and* $t_\alpha$ *denotes the* $(1-\alpha)$-*quantile of the* $t$-*distribution with* $(N-1)$ *degrees of freedom.*

*Proof.* Let $q_{\boldsymbol{\phi}}(\boldsymbol{x}) \triangleq \arg\max_{c \in \mathcal{Y}} p_{\boldsymbol{\phi}}(c \mid \boldsymbol{x})$ denote the (smoothed) diffusion classifier. For each verification sample $j \in [N]$ and noise draw $i \in [M]$, define

$$
\begin{aligned}
E_{\mathrm{sus}}^{(j,i)} &\triangleq \mathbb{I}\left\{ q_{\boldsymbol{\phi}}\left( \boldsymbol{x}_{\boldsymbol{\theta}_{\mathrm{sus}}+\boldsymbol{\epsilon}_k^{(i)}}^{(j)} \right) = \tilde{y} \right\}, \\
E_{\mathrm{ref}}^{(j,i)} &\triangleq \mathbb{I}\left\{ q_{\boldsymbol{\phi}}\left( \boldsymbol{x}_{\boldsymbol{\theta}_{\mathrm{ref}}+\boldsymbol{\epsilon}_k^{(i)}}^{(j)} \right) = \tilde{y} \right\}.
\end{aligned}
\quad (A.2)
$$

For fixed $(j, i)$, these are Bernoulli variables with means

$$
\begin{aligned}
p_{\mathrm{sus}} &\triangleq \Pr[q_{\boldsymbol{\phi}}(\boldsymbol{x}_{\boldsymbol{\theta}_{\mathrm{sus}}+\boldsymbol{\epsilon}_k}) = \tilde{y}], \\
p_{\mathrm{ref}} &\triangleq \Pr[q_{\boldsymbol{\phi}}(\boldsymbol{x}_{\boldsymbol{\theta}_{\mathrm{ref}}+\boldsymbol{\epsilon}_k}) = \tilde{y}].
\end{aligned}
\quad (A.3)
$$

For each $j$, define the noise-averaged scores

$$
\begin{aligned}
\widehat{\mathrm{WR}}_j &\triangleq \frac{1}{M} \sum_{i=1}^M E_{\mathrm{sus}}^{(j,i)}, \\
\widehat{\mathrm{RP}}_j &\triangleq \frac{1}{M} \sum_{i=1}^M E_{\mathrm{ref}}^{(j,i)}.
\end{aligned}
\quad (A.4)
$$

Therefore,

$$\mathrm{WR} = \frac{1}{N} \sum_{j=1}^N \widehat{\mathrm{WR}}_j, \qquad \mathrm{RP} = \frac{1}{N} \sum_{j=1}^N \widehat{\mathrm{RP}}_j. \quad (A.5)$$

Define paired differences $D_j \triangleq \widehat{\mathrm{WR}}_j - \widehat{\mathrm{RP}}_j$ and $\bar{D} \triangleq \frac{1}{N} \sum_{j=1}^N D_j = \mathrm{WR} - \mathrm{RP}$. The paired $t$-test uses

$$
\begin{aligned}
T &\triangleq \frac{\sqrt{N}\,\bar{D}}{s_D} \approx t(N-1), \\
s_D^2 &\triangleq \frac{1}{N-1} \sum_{j=1}^N (D_j - \bar{D})^2.
\end{aligned}
\quad (A.6)
$$

Expanding $s_D^2$ gives

$$s_D^2 = \frac{1}{N-1} \sum_{j=1}^{N} \left[ (\widehat{\mathrm{WR}}_j - \mathrm{WR}) - (\widehat{\mathrm{RP}}_j - \mathrm{RP}) \right]^2. \tag{A.7}$$

Since $\widehat{\mathrm{WR}}_j$ and $\widehat{\mathrm{RP}}_j$ are averages of $M$ Bernoulli variables, we use the standard plug-in approximation

$$\begin{aligned} \mathrm{Var}(\widehat{\mathrm{WR}}_j) &\approx \frac{1}{M} \, \mathrm{WR} \, \overline{\mathrm{WR}}, \\ \mathrm{Var}(\widehat{\mathrm{RP}}_j) &\approx \frac{1}{M} \, \mathrm{RP} \, \overline{\mathrm{RP}}, \end{aligned} \tag{A.8}$$

where $\overline{\mathrm{WR}} \triangleq 1 - \mathrm{WR}$ and $\overline{\mathrm{RP}} \triangleq 1 - \mathrm{RP}$. Accordingly, we approximate the sample variance of $\{D_j\}_{j=1}^{N}$ by

$$s_D^2 \approx \frac{1}{M} \left( \mathrm{WR} \, \overline{\mathrm{WR}} + \mathrm{RP} \, \overline{\mathrm{RP}} \right). \tag{A.9}$$

To reject $H_0$ at significance level $\alpha$, we require $T > t_\alpha$, *i.e.*,

$$\frac{\sqrt{N} \, (\mathrm{WR} - \mathrm{RP})}{s_D} > t_\alpha. \tag{A.10}$$

Substituting $s_D$ and rearranging yields

$$\sqrt{MN} \, (\mathrm{WR} - \mathrm{RP}) - t_\alpha \sqrt{\mathrm{WR} \, \overline{\mathrm{WR}} + \mathrm{RP} \, \overline{\mathrm{RP}}} > 0. \tag{A.11}$$

We next derive the closed-form threshold on WR under a known upper bound $\mathrm{RP} \le \zeta$.

$$\sqrt{MN} \, (\mathrm{WR} - \zeta) - t_\alpha \sqrt{\mathrm{WR} \, \overline{\mathrm{WR}} + \zeta(1-\zeta)} > 0. \tag{A.12}$$

Under the above conservative rule, declaring ownership is guaranteed whenever (A.12) holds.

Since $\sqrt{MN} > 0$, we square both sides of (A.12) (noting that it already enforces $\mathrm{WR} > \zeta$) and obtain

$$MN(\mathrm{WR} - \zeta)^2 > t_\alpha^2 \Big( \mathrm{WR}(1 - \mathrm{WR}) + \zeta(1-\zeta) \Big). \tag{A.13}$$

Expanding and rearranging yields a quadratic inequality:

$$\begin{aligned} &\Big( MN + t_\alpha^2 \Big) \mathrm{WR}^2 - \Big( 2MN\zeta + t_\alpha^2 \Big) \mathrm{WR} \\ &+ \Big( MN\zeta^2 - t_\alpha^2\zeta + t_\alpha^2\zeta^2 \Big) > 0. \end{aligned} \tag{A.14}$$

Let $f(\mathrm{WR})$ denote the left-hand side of (A.14). Since $MN + t_\alpha^2 > 0$, $f$ is an upward parabola. Moreover,

$$f(\zeta) = 2t_\alpha^2 \zeta(\zeta - 1) < 0, \qquad \text{for } \zeta \in (0,1), \tag{A.15}$$

and

$$f(1) = (1 - \zeta)\Big( MN(1 - \zeta) - t_\alpha^2\zeta \Big). \tag{A.16}$$

In regimes where $MN(1 - \zeta) > t_\alpha^2\zeta$, we have $f(1) > 0$. Therefore, $f$ admits a unique root $\mathrm{WR}_2 \in (\zeta, 1)$, and $f(\mathrm{WR}) > 0$ holds iff $\mathrm{WR} > \mathrm{WR}_2$ over $\mathrm{WR} \in [\zeta, 1]$.

Applying the quadratic formula to (A.14) and selecting the larger root yields

$$\mathrm{WR} > \frac{2MN\zeta + t_\alpha^2 + \sqrt{\Gamma}}{2\big(MN + t_\alpha^2\big)}, \tag{A.17}$$

where $\Gamma = \big(2MN\zeta + t_\alpha^2\big)^2 - 4\big(MN + t_\alpha^2\big)\big(MN\zeta^2 - t_\alpha^2\zeta + t_\alpha^2\zeta^2\big)$. $\qquad \square$

### A.2. Proof of Theorem 4.9

In this section, we prove Theorem A.7, which establishes certified robustness guarantees for the ownership verification of Cert-LAS. Since our layer-adaptive allocation assigns different noise levels across layers, a uniform $\ell_2$-ball fails to capture the geometry of admissible perturbations, motivating the Mahalanobis-based neighborhood in Definition A.2. Before presenting the proof, we first introduce Definition A.3 and Lemmas A.4–A.6 as preliminaries.

**Definition A.2** (Mahalanobis Ellipsoidal Neighborhood)**.** Let $\boldsymbol{\sigma}_k = (k\sigma_1, \ldots, k\sigma_L)$ denote the layerwise standard deviations under the Gaussian smoothing in Remark 4.3, and let $\|\boldsymbol{\delta}\|_{\boldsymbol{\sigma}_k} \triangleq \big(\sum_{l=1}^{L} \frac{\|\boldsymbol{\delta}^l\|_2^2}{\sigma_{k,l}^2}\big)^{1/2}$ be the induced Mahalanobis norm. The ellipsoidal neighborhood of $\boldsymbol{\theta}$ with radius $R_k$ is defined as:

$$\mathcal{B}_{\boldsymbol{\sigma}_k}(\boldsymbol{\theta}; R_k) \triangleq \big\{ \boldsymbol{\theta} + \boldsymbol{\delta} \in \mathbb{R}^d \mid \|\boldsymbol{\delta}\|_{\boldsymbol{\sigma}_k} \le R_k \big\}, \tag{A.18}$$

where $R_k$ bounds the $\boldsymbol{\sigma}_k$-weighted magnitude of admissible perturbations. Since $\boldsymbol{\sigma}_k = k\boldsymbol{\sigma}_1$, we define the normalized radius $\bar{R} \triangleq R_1$, so that $R_k = \bar{R}/k$ for any $k > 0$.

**Definition A.3** (Type-I/II Error in Model Watermark Detection)**.** Let $X$ be a random variable taking values in $\mathcal{X}$, with distribution $\mathbb{P}_0$ under the null hypothesis $H_0$ (*i.e.*, the suspected generator $G(\cdot; \boldsymbol{\theta}_{\mathrm{sus}})$ does not contain watermark $\mathcal{W}$) and distribution $\mathbb{P}_1$ under the alternative hypothesis $H_1$ (*i.e.*, $G(\cdot; \boldsymbol{\theta}_{\mathrm{sus}})$ contains $\mathcal{W}$). For a sample $\boldsymbol{x} \in \mathcal{X}$, a randomized test $\psi : \mathcal{X} \to [0, 1]$ specifies the probability of rejecting $H_0$. The Type-I and Type-II errors are defined as follows:

- **Type-I Error** ($\beta_1$): The probability of incorrectly identifying an unwatermarked model as watermarked (*i.e.*, $H_0$ is true but rejected):

$$\beta_1(\psi; \mathbb{P}_0) = \mathbb{E}_{\mathbb{P}_0}[\psi(\boldsymbol{x})]. \tag{A.19}$$

- **Type-II Error** ($\beta_2$): The probability of incorrectly identifying a watermarked model as unwatermarked (*i.e.*, $H_0$ is false but accepted):

$$\beta_2(\psi; \mathbb{P}_1) = \mathbb{E}_{\mathbb{P}_1}[1 - \psi(\boldsymbol{x})]. \tag{A.20}$$

In model ownership verification, Type-I errors mistakenly flag an unwatermarked model as infringing, while Type-II errors allow adversarially modified models to evade detection. Since watermark detection often serves as a preliminary step before legal proceedings, provable bounds on false positives are essential for evidential admissibility. We note that the Type-I/II errors defined here are instantiated in the MOV context with $(\mathbb{P}_0, \mathbb{P}_1)$ specifying the unwatermarked and watermarked hypotheses; in the subsequent analysis (*e.g.*, Lemmas A.4–A.5), the same notation $\beta_1(\cdot)$ and $\beta_2(\cdot)$ refers to the generic Type-I/II errors from the Neyman-Pearson framework, where $(\mathbb{P}_0, \mathbb{P}_1)$ may correspond to other distribution pairs such as $(Z, Z + \boldsymbol{\delta})$ under smoothing noise.

Accordingly, inspired by the optimal likelihood ratio test $\psi^*$ established by the Neyman-Pearson lemma (Neyman & Pearson, 1933), we seek to maximize verification power subject to a controlled false positive rate. Let $\alpha$ denote the significance level specifying the maximum tolerable false positive rate. The optimal test $\psi^*$ then satisfies:

$$\beta_1(\psi^*; \mathbb{P}_0) = \alpha, \qquad \beta_2(\psi^*; \mathbb{P}_1) = \beta_2^*(\alpha; \mathbb{P}_0, \mathbb{P}_1), \tag{A.21}$$

where $\beta_2^*(\alpha; \mathbb{P}_0, \mathbb{P}_1) = \inf_{\psi: \beta_1(\psi; \mathbb{P}_0) \leq \alpha} \beta_2(\psi; \mathbb{P}_1)$.

**Lemma A.4.** *(Weber et al., 2023) Let $X_0$ and $X_1$ be two random variables with densities $f_0$ and $f_1$ with respect to a measure $\mu$ and denote by $\Lambda$ the likelihood ratio $\Lambda(x) = f_1(x)/f_0(x)$. For $p \in [0, 1]$ let $t_p \triangleq \inf\{t \geq 0\colon \mathbb{P}(\Lambda(X_0) \leq t) \geq p\}$. Then it holds that*

$$\mathbb{P}(\Lambda(X_0) < t_p) \leq p \leq \mathbb{P}(\Lambda(X_0) \leq t_p). \tag{A.22}$$

**Lemma A.5.** *(Weber et al., 2023) Let $X_0$ and $X_1$ be random variables taking values in $\mathcal{Z}$ and with probability density functions $f_0$ and $f_1$ with respect to a measure $\mu$. Let $\varphi^*$ be a likelihood ratio test for testing the null $X_0$ against the alternative $X_1$. Then for any deterministic function $\varphi\colon \mathcal{Z} \to [0, 1]$ the following implications hold:*

*i)* $\beta_1(\varphi) \geq 1 - \beta_1(\varphi^*) \Rightarrow 1 - \beta_2(\varphi) \geq \beta_2(\varphi^*)$

*ii)* $\beta_1(\varphi) \leq \beta_1(\varphi^*) \Rightarrow \beta_2(\varphi) \geq \beta_2(\varphi^*)$.

Building on Definition A.3, Eq. (A.21), and Lemmas A.4–A.5 (Weber et al., 2023), and inspired by Kumar et al. (2020), which certifies the stability of continuous probabilistic outputs rather than discrete Top-1 predictions, we derive a general robustness condition for Cert-LAS.

**Lemma A.6** (General Layer-Adaptive Robustness Condition). *Consider layer-adaptive smoothing noise $\mathcal{E}_k$ and perturbation $\boldsymbol{\delta} = (\boldsymbol{\delta}^1, \ldots, \boldsymbol{\delta}^L)$. For probability thresholds $a \leq s_1 \leq \cdots \leq s_m \leq b$, let $P_{s_j}(\boldsymbol{\theta}) \leq \mathbb{P}_{\boldsymbol{\epsilon}_k \sim \mathcal{E}_k}\left[\frac{1}{N}\sum_{i=1}^N \mathbb{I}\left\{q_\phi\left(\boldsymbol{x}_{\boldsymbol{\theta}+\boldsymbol{\epsilon}_k}^{(i)}\right) = \tilde{y}\right\} \geq s_j\right]$. Diffusion*

*model ownership verification is certified robust against $\boldsymbol{\delta}$ if*

$$a + (s_1 - a)\,\beta_2(1 - P_{s_1}(\boldsymbol{\theta}); \mathbb{P}_0, \mathbb{P}_1)$$
$$+ \sum_{j=2}^m (s_j - s_{j-1})\,\beta_2\big(1 - P_{s_j}(\boldsymbol{\theta}); \mathbb{P}_0, \mathbb{P}_1\big) > \tau_{\alpha,\zeta}, \tag{A.23}$$

*where $\mathbb{P}_0$ and $\mathbb{P}_1$ are the distributions under $H_0 : Z \triangleq \boldsymbol{\theta} + \boldsymbol{\epsilon}_k \sim \mathbb{P}_0$ against $H_1 : Z + \boldsymbol{\delta} \sim \mathbb{P}_1$, and $\tau_{\alpha,\zeta}$ is the verification threshold in Theorem A.1.*

*Proof.* Let $Z \triangleq \boldsymbol{\theta} + \boldsymbol{\epsilon}_k \sim \mathbb{P}_0$ and $Z' \triangleq Z + \boldsymbol{\delta} \sim \mathbb{P}_1$, and denote by $\Lambda(z) \triangleq \frac{f_{Z'}(z)}{f_Z(z)}$ the likelihood ratio between $\mathbb{P}_1$ and $\mathbb{P}_0$. For any $p \in [0, 1]$, define $t_p \triangleq \inf\{t \geq 0 : \mathbb{P}(\Lambda(Z) \leq t) \geq p\}$ and

$$q_p = \begin{cases} 0, & \text{if } \mathbb{P}(\Lambda(Z) = t_p) = 0, \\ \frac{\mathbb{P}(\Lambda(Z) \leq t_p) - p}{\mathbb{P}(\Lambda(Z) = t_p)}, & \text{otherwise.} \end{cases} \tag{A.24}$$

By Lemma A.4, we have $\mathbb{P}(\Lambda(Z) \leq t_p) \geq p$ and $\mathbb{P}(\Lambda(Z) < t_p) \leq p$, hence $q_p \in [0, 1]$. Define the likelihood ratio test

$$\varphi_p(z) = \begin{cases} 1, & \text{if } \Lambda(z) > t_p, \\ q_p, & \text{if } \Lambda(z) = t_p, \\ 0, & \text{if } \Lambda(z) < t_p. \end{cases} \tag{A.25}$$

Then $\varphi_p$ has type-I error $\beta_1(\varphi_p) = \mathbb{P}_0(\varphi_p = 1) = 1 - p$.

Fix thresholds $a \leq s_1 \leq \cdots \leq s_m \leq b$. For each $j \in [m]$, let $\varphi_{s_j} \equiv \varphi_{P_{s_j}(\boldsymbol{\theta})}$ so that

$$\beta_1(\varphi_{s_j}) = 1 - P_{s_j}(\boldsymbol{\theta}). \tag{A.26}$$

For each threshold $s \in [0, 1]$, define the events

$$A_s \triangleq \left\{\frac{1}{N}\sum_{i=1}^N \mathbb{I}\left\{q_\phi\left(\boldsymbol{x}_{\boldsymbol{\theta}+\boldsymbol{\epsilon}_k}^{(i)}\right) = \tilde{y}\right\} \geq s\right\},$$
$$B_s \triangleq \left\{\frac{1}{N}\sum_{i=1}^N \mathbb{I}\left\{q_\phi\left(\boldsymbol{x}_{\boldsymbol{\theta}+\boldsymbol{\delta}+\boldsymbol{\epsilon}_k}^{(i)}\right) = \tilde{y}\right\} \geq s\right\}. \tag{A.27}$$

By the definition of $P_s(\boldsymbol{\theta})$, we have

$$\mathbb{P}_0(A_{s_j}) = \mathbb{P}_{\boldsymbol{\epsilon}_k \sim \mathcal{E}_k}(A_{s_j}) \geq P_{s_j}(\boldsymbol{\theta}), \qquad \forall j \in [m]. \tag{A.28}$$

Consider the deterministic function $\varphi(z) \triangleq \mathbb{I}\{A_{s_j}\}$ and the likelihood ratio test $\varphi^* \equiv \varphi_{s_j}$. Since $\beta_1(\varphi) = \mathbb{P}_0(\varphi = 1) = \mathbb{P}_0(A_{s_j}) \geq P_{s_j}(\boldsymbol{\theta}) = 1 - \beta_1(\varphi_{s_j})$ and $\varphi^*$ yields

$$\mathbb{P}_1(B_{s_j}) = 1 - \beta_2(\varphi) \geq \beta_2(\varphi_{s_j})$$
$$= \beta_2\big(1 - P_{s_j}(\boldsymbol{\theta}); \mathbb{P}_0, \mathbb{P}_1\big), \forall j \in [m]. \tag{A.29}$$

Next, define the random variable

$$U \triangleq \frac{1}{N}\sum_{i=1}^N \mathbb{I}\left\{q_\phi\left(\boldsymbol{x}_{\boldsymbol{\theta}+\boldsymbol{\delta}+\boldsymbol{\epsilon}_k}^{(i)}\right) = \tilde{y}\right\} \in [a, b]. \tag{A.30}$$

We lower bound $\mathbb{E}_{\mathbb{P}_1}[U]$ by a Riemann-sum decomposition over the bins $[a, s_1), [s_1, s_2), \dots, [s_m, b]$. Using $U \geq a$ on $B_{s_1}^c$, $U \geq s_1$ on $B_{s_1} \setminus B_{s_2}$, ..., and $U \geq s_m$ on $B_{s_m}$, taking expectations under $\mathbb{P}_1$ yields

$$\mathbb{E}_{\mathbb{P}_1}[U] \geq a + (s_1 - a)\mathbb{P}_1(B_{s_1}) + \sum_{j=2}^{m}(s_j - s_{j-1})\mathbb{P}_1(B_{s_j}). \tag{A.31}$$

Substituting the bound (A.29) into the above yields

$$\begin{aligned} \mathbb{E}_{\mathbb{P}_1}[U] &\geq a + (s_1 - a)\,\beta_2(1 - P_{s_1}(\boldsymbol{\theta}); \mathbb{P}_0, \mathbb{P}_1) \\ &\quad + \sum_{j=2}^{m}(s_j - s_{j-1})\,\beta_2\big(1 - P_{s_j}(\boldsymbol{\theta}); \mathbb{P}_0, \mathbb{P}_1\big). \end{aligned} \tag{A.32}$$

Finally, ownership verification under perturbation $\boldsymbol{\delta}$ follows if the watermark robustness exceeds the verification threshold $\tau_{\alpha,\zeta}$ in Theorem A.1. Therefore, the condition

$$\begin{aligned} &a + (s_1 - a)\,\beta_2(1 - P_{s_1}(\boldsymbol{\theta}); \mathbb{P}_0, \mathbb{P}_1) \\ &+ \sum_{j=2}^{m}(s_j - s_{j-1})\,\beta_2\big(1 - P_{s_j}(\boldsymbol{\theta}); \mathbb{P}_0, \mathbb{P}_1\big) > \tau_{\alpha,\zeta} \end{aligned} \tag{A.33}$$

implies $\mathbb{E}_{\mathbb{P}_1}[U] > \tau_{\alpha,\zeta}$ and thus guarantees successful verification at significance level $\alpha$. $\qquad\square$

The general robustness condition in Lemma A.6 applies to any layer-adaptive smoothing distribution. We now instantiate it with Gaussian smoothing.

**Theorem A.7** (Certified Radius under Layer-Adaptive Gaussian Smoothing). *Consider Gaussian smoothing noise $\mathcal{E}_k = \mathcal{N}(\mathbf{0}, \boldsymbol{\Sigma}_k)$ and the normalized radius $\bar{R}$ as in Definition A.2. For probability thresholds $a \leq s_1 \leq \dots \leq s_m \leq b$, let $P_{s_j}(\boldsymbol{\theta}) \leq \mathbb{P}_{\boldsymbol{\epsilon}_k \sim \mathcal{E}_k}\left[\frac{1}{N}\sum_{i=1}^{N}\mathbb{I}\left\{q_\phi\left(\boldsymbol{x}_{\boldsymbol{\theta}+\boldsymbol{\epsilon}_k}^{(i)}\right) = \tilde{y}\right\} \geq s_j\right]$. For any perturbation $\boldsymbol{\delta} \in \mathcal{B}_{\boldsymbol{\sigma}_k}(\boldsymbol{\theta}; R_k)$, diffusion model ownership verification with layer-adaptive noise is guaranteed if $\bar{R} \leq R^*$, which is obtained by solving*

$$\begin{aligned} &a + (s_1 - a)\,\Phi\left(\Phi^{-1}(P_{s_1}(\boldsymbol{\theta})) - \frac{\bar{R}}{k}\right) \\ &+ \sum_{j=2}^{m}(s_j - s_{j-1})\,\Phi\left(\Phi^{-1}\big(P_{s_j}(\boldsymbol{\theta})\big) - \frac{\bar{R}}{k}\right) > \tau_{\alpha,\zeta}, \end{aligned} \tag{A.34}$$

*where $\Phi$ is the standard Gaussian CDF, $\alpha$ is the significance level and $\tau_{\alpha,\zeta} = \frac{2MN\zeta + t_\alpha^2 + \sqrt{\Gamma}}{2\big(MN + t_\alpha^2\big)}$ is the verification threshold in Theorem A.1.*

*Proof of Theorem A.7.* We instantiate Lemma A.6 for the Gaussian layer-adaptive smoothing noise $\mathcal{E}_k = \mathcal{N}(\mathbf{0}, \boldsymbol{\Sigma}_k)$ and derive the certified condition as follows. Let

$$Z \triangleq \boldsymbol{\theta} + \boldsymbol{\epsilon}_k, \qquad Z' \triangleq Z + \boldsymbol{\delta}, \tag{A.35}$$

where $\boldsymbol{\epsilon}_k \sim \mathcal{N}(\mathbf{0}, \boldsymbol{\Sigma}_k)$ and

$$\boldsymbol{\Sigma}_k = \text{diag}(\sigma_{k,1}^2 \mathbf{I}_{d_1}, \dots, \sigma_{k,L}^2 \mathbf{I}_{d_L}), \qquad \sigma_{k,l} \triangleq k\sigma_l. \tag{A.36}$$

Denote by $\mathbb{P}_0$ and $\mathbb{P}_1$ the distributions of $Z$ and $Z'$ under the layer-adaptive smoothing, respectively.

For $z \in \mathbb{R}^{\sum_l d_l}$, the likelihood ratio $\Lambda(z) \triangleq \frac{f_{Z'}(z)}{f_Z(z)}$ satisfies

$$\begin{aligned} \log \Lambda(z) &= \langle z - \boldsymbol{\theta}, \boldsymbol{\delta} \rangle_{\boldsymbol{\Sigma}_k^{-1}} - \frac{1}{2}\langle \boldsymbol{\delta}, \boldsymbol{\delta} \rangle_{\boldsymbol{\Sigma}_k^{-1}}, \\ \langle a, b \rangle_{\boldsymbol{\Sigma}_k^{-1}} &\triangleq a^\top \boldsymbol{\Sigma}_k^{-1} b. \end{aligned} \tag{A.37}$$

By definition of $\boldsymbol{\Sigma}_k$, we have

$$\langle \boldsymbol{\delta}, \boldsymbol{\delta} \rangle_{\boldsymbol{\Sigma}_k^{-1}} = \sum_{l=1}^{L}\frac{\|\boldsymbol{\delta}^l\|_2^2}{\sigma_{k,l}^2} = \sum_{l=1}^{L}\frac{\|\boldsymbol{\delta}^l\|_2^2}{(k\sigma_l)^2} = \|\boldsymbol{\delta}\|_{\boldsymbol{\sigma}_k}^2. \tag{A.38}$$

Since the Gaussian distribution is continuous, the likelihood ratio test for testing $\mathbb{P}_0$ against $\mathbb{P}_1$ takes the form

$$\varphi_t(z) = \mathbb{I}\{\Lambda(z) \geq t\}. \tag{A.39}$$

For any $p \in [0, 1]$, choose $t_p$ such that the test $\varphi_{t_p}$ attains type-I error $1 - p$ under $\mathbb{P}_0$. Then

$$t_p = \exp\left(\Phi^{-1}(p)\|\boldsymbol{\delta}\|_{\boldsymbol{\sigma}_k} - \frac{1}{2}\|\boldsymbol{\delta}\|_{\boldsymbol{\sigma}_k}^2\right), \tag{A.40}$$

and consequently the corresponding type-II probability of $\varphi_{t_p}$ under $\mathbb{P}_1$ is

$$\beta_2(1 - p; \mathbb{P}_0, \mathbb{P}_1) = \Phi\big(\Phi^{-1}(p) - \|\boldsymbol{\delta}\|_{\boldsymbol{\sigma}_k}\big), \tag{A.41}$$

where $\Phi$ is the standard Gaussian CDF.

For each $j \in [m]$, applying Lemma A.6 with (A.41) gives

$$\begin{aligned} &\mathbb{E}_{\mathbb{P}_1}\left[\frac{1}{N}\sum_{i=1}^{N}\mathbb{I}\left\{q_\phi\left(\boldsymbol{x}_{\boldsymbol{\theta}+\boldsymbol{\delta}+\boldsymbol{\epsilon}_k}^{(i)}\right) = \tilde{y}\right\}\right] \\ &\geq a + (s_1 - a)\,\Phi\big(\Phi^{-1}(P_{s_1}(\boldsymbol{\theta})) - \|\boldsymbol{\delta}\|_{\boldsymbol{\sigma}_k}\big) \\ &+ \sum_{j=2}^{m}(s_j - s_{j-1})\,\Phi\big(\Phi^{-1}\big(P_{s_j}(\boldsymbol{\theta})\big) - \|\boldsymbol{\delta}\|_{\boldsymbol{\sigma}_k}\big). \end{aligned} \tag{A.42}$$

By Theorem A.1, verification succeeds whenever the watermark robustness rate exceeds the threshold $\tau_{\alpha,\zeta}$. Therefore, it suffices that the right-hand side of (A.42) satisfies

$$\begin{aligned} &a + (s_1 - a)\,\Phi\big(\Phi^{-1}(P_{s_1}(\boldsymbol{\theta})) - \|\boldsymbol{\delta}\|_{\boldsymbol{\sigma}_k}\big) \\ &+ \sum_{j=2}^{m}(s_j - s_{j-1})\,\Phi\big(\Phi^{-1}\big(P_{s_j}(\boldsymbol{\theta})\big) - \|\boldsymbol{\delta}\|_{\boldsymbol{\sigma}_k}\big) > \tau_{\alpha,\zeta}. \end{aligned} \tag{A.43}$$

Finally, for $\boldsymbol{\delta} \in \mathcal{B}_{\boldsymbol{\sigma}_k}(\boldsymbol{\theta}; R_k)$, Definition A.2 implies

$$\|\boldsymbol{\delta}\|_{\boldsymbol{\sigma}_k} \leq R_k = \bar{R}/k. \tag{A.44}$$

Since $\Phi(\cdot)$ is strictly increasing, each term $\Phi(\Phi^{-1}(P_{s_j}(\boldsymbol{\theta})) - r)$ is strictly decreasing in $r \geq 0$. Hence the left-hand side of (A.43) is strictly decreasing in $\|\boldsymbol{\delta}\|_{\boldsymbol{\sigma}_k}$, and it is sufficient to enforce (A.43) at the worst case $\|\boldsymbol{\delta}\|_{\boldsymbol{\sigma}_k} = \bar{R}/k$. Therefore, ownership verification is guaranteed for all perturbations with $\bar{R} \leq R^*$, where $R^*$ is the maximal radius obtained by solving (A.34). $\qquad\square$

### A.3. Proof of Tightness

In this section, we derive a theorem that establishes tightness: any perturbation outside (A.23) admits a classifier that leads to unreliable verification.

**Theorem A.8** (Tightness). *For any perturbation $\boldsymbol{\delta}$ that violates (A.23), there exists a diffusion classifier $h^*$ consistent with the probability bounds $P_{s_j}(\boldsymbol{\theta})$ for which (A.1) does not hold. Consequently, ownership verification cannot be guaranteed at significance level $\alpha$.*

*Proof.* We demonstrate the tightness of our robustness condition by constructing a worst-case base classifier $h^*$ such that its smoothed diffusion classifier fails to maintain reliable verification when condition (A.23) is violated. The key insight is to design $h^*$ to precisely achieve the lower bound in Lemma A.6, thereby representing the most challenging scenario for ownership verification.

Let

$$Z \triangleq \boldsymbol{\theta} + \boldsymbol{\epsilon}_k \sim \mathbb{P}_0, \qquad Z' \triangleq \boldsymbol{\theta} + \boldsymbol{\delta} + \boldsymbol{\epsilon}_k \sim \mathbb{P}_1, \quad \text{(A.45)}$$

and define the likelihood ratio

$$\Lambda(z) \triangleq \frac{f_{Z'}(z)}{f_Z(z)}. \quad \text{(A.46)}$$

For any $p \in [0, 1]$, let

$$t_p \triangleq \inf\left\{ t \geq 0 : \mathbb{P}\big(\Lambda(Z) \leq t\big) \geq p \right\} \quad \text{(A.47)}$$

as in Lemma A.4. For each $j = 1, 2, \ldots, m$, we further introduce the following notation:

$$T_j \triangleq t_{P_{s_j}(\boldsymbol{\theta})}. \quad \text{(A.48)}$$

Then, the base classifier $h^*$ is defined by

$$h^*(z) = \begin{cases} s_m, & \text{if } \Lambda(z) < T_m, \\ s_{m-1}, & \text{if } T_m < \Lambda(z) < T_{m-1}, \\ \vdots \\ s_1, & \text{if } T_2 < \Lambda(z) < T_1, \\ a, & \text{if } \Lambda(z) > T_1. \end{cases} \quad \text{(A.49)}$$

This construction is deliberately chosen to make verification as difficult as possible while still satisfying the given constraints: on each likelihood-ratio interval, $h^*$ assigns the minimum admissible value.

We first verify that this construction satisfies the constraints. For each $j \in [m]$, the event $\{h^*(Z) \geq s_j\}$ coincides with the acceptance region of the likelihood ratio test with threshold $T_j$, namely $\{\Lambda(Z) < T_j\}$ together with a randomized assignment on the boundary $\{\Lambda(Z) = T_j\}$. Hence,

$$\mathbb{P}_0(h^*(Z) \geq s_j) = \mathbb{P}_0(\Lambda(Z) < T_j) + (1 - q_j)\,\mathbb{P}_0(\Lambda(Z) = T_j), \quad \text{(A.50)}$$

where $q_j$ is the tie-breaking coefficient of the likelihood ratio test at threshold $T_j$:

$$q_j = \begin{cases} 0, & \text{if } \mathbb{P}_0(\Lambda(Z) = T_j) = 0, \\ \frac{\mathbb{P}_0(\Lambda(Z) \leq T_j) - P_{s_j}(\boldsymbol{\theta})}{\mathbb{P}_0(\Lambda(Z) = T_j)}, & \text{otherwise.} \end{cases} \quad \text{(A.51)}$$

By Lemma A.4, we have

$$\mathbb{P}_0(\Lambda(Z) < T_j) \leq P_{s_j}(\boldsymbol{\theta}) \leq \mathbb{P}_0(\Lambda(Z) \leq T_j). \quad \text{(A.52)}$$

Therefore, the above choice of $q_j$ ensures

$$\mathbb{P}_0(h^*(Z) \geq s_j) = P_{s_j}(\boldsymbol{\theta}). \quad \text{(A.53)}$$

Therefore, $h^*$ is consistent with the prescribed bounds $\{P_{s_j}(\boldsymbol{\theta})\}_{j=1}^m$ and qualifies as a valid classifier.

The crucial step is to show that this construction achieves the theoretical lower bound. Let $U^* \triangleq h^*(Z')$. By expanding the expectation over the disjoint regions in (A.49), we obtain

$$\mathbb{E}_{\mathbb{P}_1}[U^*] = a \cdot \mathbb{P}_1(\Lambda(Z') > T_1) \\ + \sum_{j=2}^m s_{j-1} \cdot \mathbb{P}_1(T_j < \Lambda(Z') < T_{j-1}) \quad \text{(A.54)} \\ + s_m \cdot \mathbb{P}_1(\Lambda(Z') < T_m),$$

with the same boundary randomization as above. For each $j$, consider the likelihood ratio test

$$\varphi_j(z) = \begin{cases} 1, & \text{if } \Lambda(z) > T_j, \\ q_j, & \text{if } \Lambda(z) = T_j, \\ 0, & \text{if } \Lambda(z) < T_j. \end{cases} \quad \text{(A.55)}$$

By construction, $\varphi_j$ has type-I error

$$\beta_1(\varphi_j) = \mathbb{P}_0(\Lambda(Z) > T_j) + q_j\,\mathbb{P}_0(\Lambda(Z) = T_j) = 1 - P_{s_j}(\boldsymbol{\theta}). \quad \text{(A.56)}$$

Therefore, by the definition of $\beta_2(\cdot; \mathbb{P}_0, \mathbb{P}_1)$,

$$\mathbb{P}_1(\Lambda(Z') < T_j) + (1 - q_j)\,\mathbb{P}_1(\Lambda(Z') = T_j) \\ = \beta_2\big(1 - P_{s_j}(\boldsymbol{\theta}); \mathbb{P}_0, \mathbb{P}_1\big). \quad \text{(A.57)}$$

Substituting (A.57) into (A.54) yields

$$\mathbb{E}_{\mathbb{P}_1}[U^*] = a + (s_1 - a)\,\beta_2(1 - P_{s_1}(\boldsymbol{\theta}); \mathbb{P}_0, \mathbb{P}_1) \\ + \sum_{j=2}^m (s_j - s_{j-1})\,\beta_2\big(1 - P_{s_j}(\boldsymbol{\theta}); \mathbb{P}_0, \mathbb{P}_1\big), \quad \text{(A.58)}$$

which exactly matches the lower bound in Lemma A.6, proving its tightness.

Finally, when condition (A.23) is violated, the right-hand side of (A.58) is at most $\tau_{\alpha,\zeta}$. This means the verification condition (A.1) does not hold for the constructed $h^*$. Consequently, model ownership verification cannot be guaranteed at significance level $\alpha$. □

# B. More Related Works

### B.1. Fingerprinting-based Diffusion Watermarking

Fingerprint-based diffusion watermarking embeds user-specific identifiers into generated images, enabling user-level attribution and deepfake tracing. Based on the embedding strategy, existing methods can be divided into two categories: latent space fingerprinting and fine-tuning (Fernandez et al., 2023; Xiong et al., 2023; Ma et al., 2024). Latent-space fingerprinting methods (Wen et al., 2023; Lei et al., 2024; Yang et al., 2024c) embed watermarks by modifying the frequency domain of latent representations without additional training. Partial fine-tuning methods embed watermarks through model adaptation, such as fine-tuning customized decoders for different users. However, latent-space fingerprinting requires strong externalization assumptions where model providers must maintain control over the generation code, while fine-tuning locates ownership signals outside the UNet backbone (*e.g.*, in detachable decoders), making both categories vulnerable to component substitution attacks (Wang et al., 2025d).

### B.2. Backdoor-based Diffusion Watermarking

Backdoor-based diffusion watermarking achieves model ownership verification (MOV) through a private trigger that activates a predefined watermark behavior during generation (Liu et al., 2023b; Zhao et al., 2023; Feng et al., 2024; Wang et al., 2025d), adopting either *synthetic triggers* such as rare tokens and semantically atypical patterns, or *concept triggers* that reuse pretrained concepts to induce a semantically mismatched watermark response. Among these, synthetic triggers remain the dominant design, as they reduce language drift and empirically exhibit stronger survivability under downstream fine-tuning, while concept triggers are less competitive because pretrained concepts are difficult to learn as triggers for mismatched image targets (Huang et al., 2024). However, the confidentiality of synthetic triggers is difficult to guarantee in practice due to their semantic atypicality, and this line of work lacks theoretical guarantees of certified robustness under adversarial model modifications, leaving it exposed to adaptive attacks. Beyond diffusion-specific designs, backdoor mechanisms have been broadly explored across CLIP (Liang et al., 2024c), vision-language models (Liang et al., 2025a;b), and

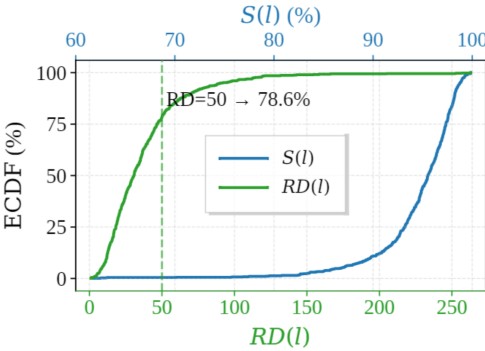

*Figure 5.* Empirical cumulative distribution functions of rank dispersion $\text{RD}(l)$ and stability score $S(l)$ across UNet layers. Most layers demonstrate low rank dispersion and high stability scores, indicating consistent update patterns across datasets.

pretrained visual backbones (Yang et al., 2024a; Liu et al., 2025), highlighting the breadth of trigger-design strategies that motivate trigger-free alternatives such as ours.

# C. Pilot Study

**Settings.** To derive a noise allocation strategy that accounts for the structure of the UNet architecture, we conduct a pilot study examining layerwise parameter updates during full fine-tuning across diverse datasets. We fine-tune Stable Diffusion v1.4 (Rombach et al., 2022) with full parameters and a learning rate of $1 \times 10^{-5}$ on 6 datasets spanning different domains: LAION-5B (Schuhmann et al., 2022), CelebA-HQ (Karras et al., 2018), Dogs vs. Cats (Elson et al., 2007), Cartoon (Norod78, 2022), Landscape (Rougetet, 2020), and ArtBench (Liao et al., 2022). For each layer $l$ in the UNet backbone, we quantify the update magnitude using the average $L_2$ norm $\bar{\delta}^l(0, T)$ as defined in Definition 4.1, measured between initialization and the final training step $T = 5000$. To assess whether these update patterns generalize across datasets, we evaluate two complementary metrics. First, we compute the rank dispersion $\text{RD}(l) = \frac{1}{|\mathcal{P}_m|} \sum_{(i,j) \in \mathcal{P}_m} |r_{l,i} - r_{l,j}|$ by ranking all layers according to their update magnitude within each dataset, then calculating the average absolute rank difference across all dataset pairs, where $r_{l,i}$ denotes the rank of layer $l$ in dataset $i$ and $\mathcal{P}_m = \{(i,j) \in [m] \times [m] | i < j\}$ is the set of unordered dataset pairs. A smaller $\text{RD}(l)$ indicates that layer $l$ maintains more consistent relative ranking across datasets. Second, we compute the stability score $S(l) = 1 - \frac{\text{RD}(l)}{L-1}$, where $L$ is the number of layers. A higher $S(l)$ indicates more consistent relative update behavior. To visualize the distribution of these metrics across all layers, we compute their empirical cumulative distribution functions (ECDF): $\widehat{F}_{\text{RD}}(t) = \frac{1}{L} \sum_{l=1}^{L} \mathbf{1}\{\text{RD}(l) \leq t\}$ and $\widehat{F}_S(s) = \frac{1}{L} \sum_{l=1}^{L} \mathbf{1}\{S(l) \leq s\}$, where $\mathbf{1}\{\cdot\}$ is the indicator function.

**Results.** As shown in Fig. 5, layerwise update patterns exhibit strong cross-dataset consistency: approximately

*Table 8.* Layers with Largest and Smallest Update Magnitude

| Top 25 Layers | Bottom 25 Layers |
| --- | --- |
| up_blocks.1.resnets.2.norm2.weight | time_embedding.linear_1.bias |
| up_blocks.2.resnets.0.norm2.weight | time_embedding.linear_2.bias |
| up_blocks.3.attentions.0.transformer_blocks.0.norm1.weight | up_blocks.0.resnets.2.time_emb_proj.weight |
| down_blocks.2.attentions.1.transformer_blocks.0.norm2.bias | up_blocks.3.attentions.2.norm.bias |
| up_blocks.0.resnets.1.norm2.weight | up_blocks.3.attentions.2.proj_in.bias |
| up_blocks.0.resnets.2.norm2.weight | up_blocks.3.resnets.2.conv2.bias |
| up_blocks.2.resnets.1.norm2.weight | down_blocks.0.resnets.0.norm1.bias |
| mid_block.resnets.1.norm2.weight | up_blocks.3.resnets.2.conv_shortcut.bias |
| up_blocks.2.resnets.2.norm2.weight | up_blocks.3.attentions.2.proj_out.bias |
| up_blocks.3.attentions.0.transformer_blocks.0.norm3.weight | down_blocks.0.resnets.0.conv2.bias |
| up_blocks.3.attentions.1.transformer_blocks.0.norm1.weight | up_blocks.3.resnets.1.conv2.bias |
| up_blocks.0.resnets.0.norm2.weight | up_blocks.0.resnets.1.time_emb_proj.weight |
| up_blocks.1.resnets.0.norm2.weight | up_blocks.3.resnets.1.conv_shortcut.bias |
| down_blocks.0.attentions.1.transformer_blocks.0.norm1.weight | up_blocks.0.resnets.0.time_emb_proj.weight |
| up_blocks.3.resnets.0.norm2.weight | mid_block.resnets.1.time_emb_proj.weight |
| conv_norm_out.bias | up_blocks.3.resnets.0.conv2.bias |
| up_blocks.3.attentions.0.transformer_blocks.0.attn2.to_k.weight | up_blocks.3.resnets.0.conv_shortcut.bias |
| up_blocks.1.attentions.1.transformer_blocks.0.norm2.weight | up_blocks.3.attentions.2.transformer_blocks.0.norm1.bias |
| up_blocks.3.attentions.0.transformer_blocks.0.norm2.weight | up_blocks.1.resnets.1.conv2.bias |
| up_blocks.1.attentions.0.transformer_blocks.0.norm2.weight | up_blocks.1.resnets.1.conv_shortcut.bias |
| up_blocks.1.resnets.1.norm2.weight | up_blocks.3.attentions.2.transformer_blocks.0.ff.net.2.bias |
| up_blocks.2.attentions.0.transformer_blocks.0.norm2.weight | mid_block.resnets.0.time_emb_proj.weight |
| up_blocks.1.attentions.2.transformer_blocks.0.norm2.weight | down_blocks.0.attentions.1.transformer_blocks.0.ff.net.2.bias |
| up_blocks.3.attentions.0.transformer_blocks.0.attn2.to_q.weight | down_blocks.0.attentions.1.proj_out.bias |
| up_blocks.2.attentions.2.transformer_blocks.0.norm2.weight | up_blocks.3.attentions.1.proj_out.bias |

78.6% of layers satisfy rank dispersion $\mathrm{RD}(l) \leq 50$ and stability scores $S(l) > 0.5$. Under a random ranking baseline, achieving such agreement would have probability $\approx 10^{-1543}$, which suggests that the observed update magnitudes may be driven more by intrinsic architectural properties than by idiosyncrasies of any particular dataset.

To shed light on this phenomenon, Tab. 8 highlights several plausible explanations. Among the top 25 layers, normalization parameters (*e.g.*, `up_blocks.*.norm2.weight`) appear frequently, which could be consistent with their role in modulating feature distributions toward dataset-specific statistics (Ronneberger et al., 2015). Moreover, upsampling blocks and cross-attention components (`attn2.to_q/k.weight`) also rank highly, potentially reflecting their involvement in reconstructing fine-grained visual details and encoding domain-dependent semantic relations (Rombach et al., 2022; Kumari et al., 2023; Bao et al., 2024). In contrast, the bottom 25 layers are dominated by time-embedding projections (*e.g.*, `time_emb_proj.weight`), bias terms, and certain linear projections. One possible interpretation is that time embeddings primarily parameterize diffusion timesteps with comparatively limited dependence on image content (Nichol & Dhariwal, 2021; Huang et al., 2023; Voynov et al., 2023), while bias terms and some linear projections may function as more generic structural parameters that require less

dataset-specific adjustment.

In summary, this asymmetry could be indicative of an efficient adaptation pattern: the model may largely preserve components related to core feature processing and noise scheduling, while preferentially adjusting modules that mediate semantic conditioning and detail synthesis. Such a mechanism could facilitate transfer across visual domains without substantially altering the model's generative behavior (Ronneberger et al., 2015; Dhariwal & Nichol, 2021). Correspondingly, the persistence of this pattern across diverse datasets motivates our $\mathrm{LFS}(l)$-guided noise allocation strategy, which prioritizes smoothing on layers that appear more susceptible to fine-tuning updates.

## D. Detailed Experimental Settings

### D.1. Model Watermarking

In the watermarking phase, we use Stable Diffusion v1.4 on the Dogs vs. Cats dataset (Elson et al., 2007) as the default setting. To derive layer-adaptive noise levels, we first perform a short fine-tuning stage with learning rate $1 \times 10^{-5}$ to compute the sensitivity indicators $\mathrm{LFS}(l)$, which then determine the layerwise noise levels $\boldsymbol{\sigma}_1$ under a global budget $\sigma_{\mathrm{u}} = 0.01$. With these noise levels established, we proceed to watermark embedding by training the generator to induce targeted misclassification.

Following prior work on diffusion-based classification (Li et al., 2023; Chen et al., 2024), we adopt a binary prompt set $\mathcal{Y} = \{$"a photo of a cat", "a photo of a dog"$\}$, designate "cat" as the watermark class prompt $\tilde{y}$, and train the generator such that samples produced under the "cat" prompt are shifted toward the "dog" side with target distribution $q^*(\lambda) = [1 - \lambda, \lambda]$, using $\lambda = 0.55$ by default. The generator is optimized with a single-step inversion-based objective, regularization weight $\omega_0 = 5 \times 10^{-5}$, and a learning rate $1 \times 10^{-6}$, yielding the watermarked generator $G(\cdot; \boldsymbol{\theta}_w)$.

### D.2. Model Verification

For watermark verification, we reuse the layer-adaptive noise levels $\{\sigma_l\}_{l=1}^{L}$ from the watermarking stage and repeatedly evaluate both the watermarked model and an unwatermarked reference model under paired parameter noise trials. Each trial produces a scalar verification statistic given by the fraction of generated images classified by the diffusion classifier as the target prompt $\tilde{y}$. To obtain finite-sample certified bounds, we instantiate the watermarked-side probabilities $P_{s_j}(\boldsymbol{\theta})$ via a one-sided Dvoretzky–Kiefer–Wolfowitz lower confidence bound, and bound the reference-side baseline rate $\zeta$ via a one-sided Hoeffding upper confidence bound. Substituting these bounds into Eq. (A.34) with $m = 100$ empirical-quantile thresholds, we determine the maximum certified radius $R^*$ by grid search.

### D.3. Effectiveness Metrics

**TPR at FPR $10^{-6}$ (T@$10^{-6}$F).** We report the true positive rate (TPR) under a stringent false positive rate (FPR) constraint of $10^{-6}$. Specifically, given images generated from verification prompts, we apply an exact Binomial sign test to determine whether the suspected model produces watermarked images while controlling the FPR at $10^{-6}$. We then report the resulting TPR as T@$10^{-6}$F. Higher values indicate stronger watermark detectability.

**Verification Success Rate (VSR).** VSR measures the proportion of verification attempts that successfully confirm model ownership under layer-adaptive smoothing. Given $M$ noise samples and $N$ verification images, VSR is calculated as the fraction of trials where the suspected model is successfully verified as watermarked. Higher values indicate more reliable ownership verification.

**Certified Radius ($\bar{R}$).** The certified radius quantifies the maximum magnitude of parameter perturbations under which ownership verification is provably guaranteed. We measure perturbations using a Mahalanobis norm induced by the layer-adaptive noise levels. A larger $\bar{R}$ indicates greater robustness against parameter modifications, providing a provable guarantee that adversaries cannot remove the watermark through bounded parameter manipulations.

**Suspiciousness Scores ($\mathcal{S}_{\text{in}}(p)$ and $\mathcal{S}_{\text{out}}(p)$).** These metrics measure the detectability of watermark triggers under adversarial auditing. The prompt suspiciousness score $\mathcal{S}_{\text{in}}(p)$ evaluates trigger detectability in the input space by computing word-level contextual incongruity using GPT-2 medium. The image suspiciousness score $\mathcal{S}_{\text{out}}(p)$ evaluates detectability in the output space by comparing the within-prompt similarity between images generated from original and de-triggered prompts. Lower scores indicate better stealthiness.

### D.4. Details of Statistical Testing

Following prior work (Wang et al., 2025d), we adopt hypothesis testing for watermark verification to ensure fair comparison across methods.

**WatermarkDM.** WatermarkDM employs hypothesis testing based on image similarity metrics. They use SSIM to measure the alignment between a generated image and the target watermark image, with the verification threshold empirically determined by evaluating SSIM scores on clean images and selecting a value that maintains the FPR below $10^{-6}$. With this threshold established, they compute T@$10^{-6}$F to assess watermark detection performance.

**SleeperMark.** SleeperMark embeds a predefined watermark message and extracts it from images generated by a suspicious model. Verification relies on counting matching bits between embedded and extracted messages: if the count exceeds a threshold, the model is deemed derived from the original. The threshold is analytically determined by assuming bits extracted from clean images follow an i.i.d. Bernoulli(0.5) distribution, which allows exact computation of the false positive rate. They set the threshold to maintain an FPR of $10^{-6}$ and average verification results across multiple triggered images to confirm ownership.

**Ours.** We employ different statistical tests for T@$10^{-6}$F and certified radius $\bar{R}$. For T@$10^{-6}$F, we apply per-sample exact sign tests, comparing classifier confidence scores between watermarked and clean models to compute exact p-values via the binomial distribution, with a sample detected if the p-value falls below $10^{-6}$. This image-level hypothesis testing aligns with baselines at the $10^{-6}$ significance level for fair comparison. We evaluate two settings: with randomized smoothing, where paired parameter noise is added to both models to test robustness under parameter perturbations, and without smoothing, where only input randomness varies. This approach effectively captures the directional consistency of subtle confidence shifts around 0.5 (e.g., 0.5001 vs. 0.4999) regardless of magnitude. For computing $\bar{R}$ via VSR, we employ a paired-sample $t$-test at significance level $\alpha = 0.05$, as our theoretical guarantee requires only the mean VSR based on whether the target class label is successfully predicted, and the averaged success rates within $[0, 1]$ satisfy the normality assumption.

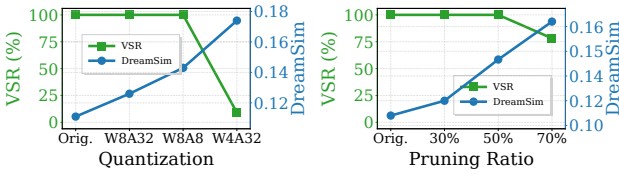
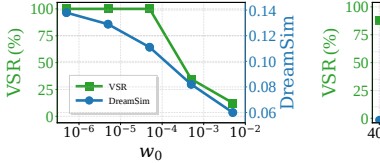
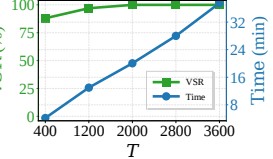

*(a)* Model Quantization          *(b)* Model Pruning

*Figure 6.* Robustness of Cert-LAS under Model Compression.

*(a)* Ablation study on $\omega_0$          *(b)* Ablation study on $T$

*Figure 7.* Impact of $\omega_0$ on VSR and DreamSim (left), and $T$ on VSR and training time (right).

*Table 9.* Training efficiency comparison on 4 A100 80GB GPUs.

| Method | Time |
| --- | --- |
| WatermarkDM | 3.2h |
| SleeperMark | 13.7h |
| Cert-LAS (w/o Exp.) | 9.4h |
| **Cert-LAS (w Exp.)** | **2.2h** |

*Table 10.* Impact of sampling budget $M$ and $N$ on verification.

| Setting | Time | VSR |
| --- | --- | --- |
| $M$=10, $N$=100 | 0.23h | 0.676 |
| $M$=100, $N$=100 | 2.22h | 0.671 |
| $M$=1000, $N$=100 | 22.17h | 0.670 |
| $M$=100, $N$=1000 | 21.10h | 0.671 |

## D.5. Details of Unintentional Attack

We evaluate watermark robustness against unintentional attacks through fine-tuning on diverse downstream tasks. For full fine-tuning, we employ four large-scale datasets spanning distinct visual domains: Cartoon (Norod78, 2022) for stylized illustrations, CelebA-HQ (Karras et al., 2018) for high-resolution facial imagery, Landscape (Rougetet, 2020) for natural scenery, and ArtBench (Liao et al., 2022) for artistic styles. Additionally, we consider three parameter-efficient fine-tuning methods that represent common adaptation scenarios: LoRA (Hu et al., 2022) fine-tuned on the Naruto-style dataset (Cervenka, 2022), DreamBooth (Ruiz et al., 2023) personalized on the Dog dataset (Ruiz et al., 2023), and Custom Diffusion (Kumari et al., 2023) trained on tortoise plushy (Kumari et al., 2023). These datasets and fine-tuning methods collectively represent typical model customization scenarios encountered in practice.

## D.6. Details of Intentional Attack

In this paper, we evaluate robustness against parameter-space attacks for three watermarking methods: WatermarkDM uses an attack that optimizes model parameters through backpropagation across the entire diffusion process to make watermarked outputs structurally similar to clean model generations, thereby reducing the SSIM score with the reference watermark image below the detection threshold; SleeperMark faces an attack that trains the model to remove the embedded watermark residual signal from its outputs, causing the watermark extractor to decode all-zero bits instead of the private binary string; and our method is attacked by training the model so that the frozen classifier correctly classifies the generated images as the true class prompt instead of the watermark class.

## E. Resistance to Model Compression

Beyond fine-tuning and parameter perturbations, a verifier may encounter suspect models that have undergone com-

pression such as *quantization* or *pruning*, which can substantially alter parameter values. We therefore evaluate whether Cert-LAS remains verifiable under these transformations.

**Settings.** For quantization, we apply post-training quantization to the watermarked model using *torchao* (Or et al., 2025), at W8A32, W8A8, and W4A32 precision. For pruning, we apply structured pruning using *OBSDiff* (Zhu et al., 2026) at sparsity ratios of 30%, 50%, and 70%. In all cases the private classifier and reference generator are kept fixed. We report the verification success rate (VSR) to measure watermark robustness under compression, together with DreamSim to quantify the degradation in generation quality.

**Results.** As shown in Fig. 6, Cert-LAS withstands both quantization and pruning unless the compression itself destroys generation quality. Specifically, Cert-LAS preserves VSR = 1.000 across moderate quantization and pruning. While VSR does drop markedly under the most aggressive W4A32 quantization, DreamSim has by then risen well above its watermarked baseline, indicating the model is no longer practically usable. This indicates that Cert-LAS exhibits strong robustness against model compression, as the watermark cannot be removed without simultaneously reducing the stolen model's generation quality.

## F. Ablation Study

### F.1. Ablation on Hyperparameters

**Impact of Initial Regularization Weight.** Fig. 7a reveals a clear trade-off between robust verification (measured by VSR) and model fidelity (measured by DreamSim, where lower indicates better preservation of visual quality) when varying the initial regularization weight $\omega_0$. Increasing $\omega_0$ improves model fidelity by reducing DreamSim, but concurrently degrades watermark effectiveness, leading to lower VSR. When $\omega_0$ falls below $5 \times 10^{-5}$, VSR remains saturated while DreamSim increases substantially, indicating

*Table 11.* Ablation on the reference generator (WR–RP gap).

*(a)* Distinct architectures and scales.

| Reference Generator | WR–RP gap |
| --- | --- |
| SD v1.4 | 0.875 |
| SDXL | 0.891 |
| Z-Image | 0.856 |

*(b)* Hardest-to-distinguish SD v1.3–v1.5 variants.

| Model | ArtBench | CelebA-HQ | Landscape |
| --- | --- | --- | --- |
| SD v1.3 | 0.864 | 0.891 | 0.888 |
| SD v1.4 | 0.876 | 0.887 | 0.880 |
| SD v1.5 | 0.893 | 0.872 | 0.899 |

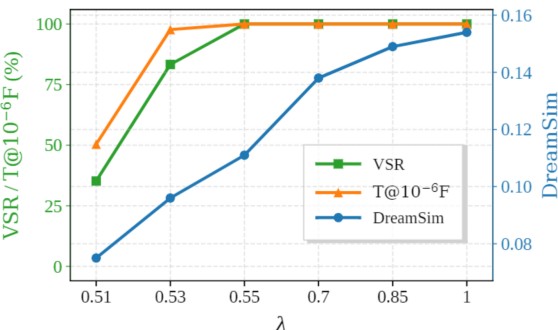

*Figure 8.* Impact of the target distribution $q^*(\lambda)$ on verification and fidelity.

that weaker regularization embeds watermarks more aggressively at the expense of generation quality. Based on these observations, $\omega_0$ of $5 \times 10^{-5}$ strikes a reasonable balance between robust verification and model fidelity.

**Impact of Sensitivity Estimation Steps and Training Schedule.** As illustrated in Fig. 7b, VSR saturates quickly while runtime scales linearly with increasing sensitivity estimation steps $T$. Specifically, even 13 to 20 minutes of sensitivity estimation suffices to achieve high VSR, indicating that reliable layer sensitivity indicators can be efficiently obtained without extensive computation. Based on this, we further explore the efficiency of the Exponential Growth Schedule proposed in Section 4.4. As shown in Tab. 9, with $\mathrm{LFS}(l, T = 2000)$ and the Exponential Growth Schedule, Cert-LAS achieves a total training time of 2.2h on 4 A100 80GB GPUs. Notably, the Exponential Growth Schedule alone reduces training time by 76.6% compared to Cert-LAS without this schedule, while also outperforming existing watermarking methods by substantial margins, suggesting that our method remains highly efficient despite the computational demands of robust training on large-scale text-to-image diffusion models.

**Impact of Sampling Budget.** In verification, $M$ denotes the number of layer-adaptive noise samples and $N$ the number of verification images. We evaluate VSR on models attacked by PGD with $\ell_2 = 0.8$, since unattacked models consistently achieve 100% VSR. As shown in Tab. 10, VSR exhibits diminishing returns with increasing sampling budget. For instance, $M = N = 100$ achieves comparable accuracy to $M = 1000$ or $N = 1000$, indicating that model owners can reliably verify ownership with a small sampling budget.

**Impact of the Target Distribution $q^*(\lambda)$.** To study the impact of the target distribution $q^*$, we parameterize it in the binary case as $q^*(\lambda) = [1 - \lambda, \lambda]$. We then embed the watermark into SD v1.4 under different values of $\lambda$, which controls the strength of the embedded watermark signal during training, and report the results in Fig. 8. The table illustrates a trade-off between verification robustness and model fidelity. Increasing $\lambda$ enhances verification but de-

grades fidelity. Notably, when $\lambda \geq 0.55$, both T@$10^{-6}$F and VSR saturate at $1.000$ with diminishing improvements, while DreamSim continues to increase as the classification loss progressively dominates the perceptual regularizer, indicating a notable decline in fidelity. This suggests that $\lambda = 0.55$ (i.e., $q^* = [0.45, 0.55]$) strikes a reasonable balance between verification robustness and model fidelity.

### F.2. Ablation on Model-Dependent Components

The WR–RP verification involves three model-dependent components: the reference generator, the suspect model, and the private classifier. We ablate each in isolation while keeping the rest of the pipeline at its default configuration. Throughout, we report the WR–RP gap, since the ownership decision depends only on whether this gap is significant: a large gap indicates reliable verification, while a collapsed gap indicates the decision threshold $\tau_{\alpha,\zeta}$ cannot be exceeded. Unless otherwise stated, RP is 0.125 under the default reference generator and private classifier.

**Reference Generator (Inference-Time).** Since RP is estimated on the reference generator, we test whether the WR–RP gap is sensitive to its choice, replacing it with **(a)** distinct architectures and scales (Stable Diffusion XL (SDXL) (Podell et al., 2023), Z-Image), and **(b)** the hardest-to-distinguish SD v1.3–v1.5 variants fine-tuned on Art-Bench, CelebA-HQ, and Landscape, which constitute a worst case as they share initialization and architecture, differing only in continued-training steps. As shown in Tab. 11, the gap stays consistently large, above 0.85 across distinct architectures and within 0.864 to 0.899 for the hardest variants. This indicates the WR–RP separation does not rely on any particular reference generator.

**Suspect Model (Inference-Time).** Using the same two settings as suspects, with the private classifier and reference generator fixed (hence RP = 0.125 and $\tau_{\alpha,\zeta} = 0.426$), we check for false attribution of non-watermarked models. As shown in Tab. 12, the gap collapses for every suspect, with WR staying far below $\tau_{\alpha,\zeta}$ and under 0.13 even for the hardest-to-distinguish variants. This suggests Cert-LAS

*Table 12.* False-positive evaluation on non-watermarked suspects (the WR–RP gap collapses, WR $< \tau_{\alpha,\zeta} = 0.426$; RP $= 0.125$).

*(a)* Distinct architectures and scales.

| Suspect Model | WR |
|---|---|
| SDXL | 0.312 |
| Z-Image | 0.008 |

*(b)* Hardest-to-distinguish SD v1.3–v1.5 variants.

| Model | ArtBench | CelebA-HQ | Landscape |
|---|---|---|---|
| SD v1.3 | 0.114 | 0.101 | 0.126 |
| SD v1.4 | 0.123 | 0.108 | 0.127 |
| SD v1.5 | 0.107 | 0.102 | 0.110 |

*Table 13.* Ablation on the watermarking-stage private classifier (WR–RP gap $> 0.79$).

| Private Classifier | WR–RP gap |
|---|---|
| SD v1.4 | 0.875 |
| SDXL | 0.790 |
| Z-Image | 0.987 |

*Table 14.* Replacing the private classifier at inference time (the WR–RP gap collapses, WR $\ll \tau_{\alpha,\zeta}$).

| Classifier (replaced) | WR | RP | $\tau_{\alpha,\zeta}$ |
|---|---|---|---|
| SD v1.4 (ArtBench) | 0.004 | 0.102 | 0.351 |
| SD v1.4 (CelebA-HQ) | 0.048 | 0.008 | 0.219 |
| SD v1.4 (Landscape) | 0.019 | 0.197 | 0.456 |
| SDXL | 0.232 | 0.210 | 0.469 |
| Z-Image | 0.001 | 0.003 | 0.233 |

*Table 15.* Cross-architecture robustness (T@$10^{-6}$F) under full fine-tuning on four datasets.

| Dataset | Backbone | 500 | 1000 | 1500 | 2000 |
|---|---|---|---|---|---|
| ArtBench | SDXL | 1.000 | 1.000 | 0.996 | 0.988 |
| | Sana-1.6B | 1.000 | 1.000 | 0.999 | 0.993 |
| CelebA-HQ | SDXL | 1.000 | 1.000 | 1.000 | 0.999 |
| | Sana-1.6B | 1.000 | 1.000 | 1.000 | 0.995 |
| Cartoon | SDXL | 1.000 | 1.000 | 1.000 | 0.990 |
| | Sana-1.6B | 1.000 | 1.000 | 1.000 | 0.995 |
| Landscape | SDXL | 1.000 | 1.000 | 1.000 | 0.998 |
| | Sana-1.6B | 1.000 | 1.000 | 1.000 | 0.996 |

does not produce false positives, independent of the suspect's architecture or pretraining background.

**Private Classifier (Watermarking-Stage).** We embed the watermark using SD v1.4, SDXL, and Z-Image as the private classifier. As shown in Tab. 13, the WR–RP gap remains above 0.79 across all architectures. This indicates that the private classifier and the protected generator can adopt different architectures during watermark embedding, granting the defender flexibility in selecting the classifier backbone.

**Private Classifier (Inference-Time).** We hereby replace the private classifier at verification stage, using SD v1.4 variants fine-tuned on ArtBench, CelebA-HQ, and Landscape, as well as SDXL and Z-Image, while the watermarking-stage classifier remains unchanged. As shown in Tab. 14, the gap collapses in all cases, with WR staying far below $\tau_{\alpha,\zeta}$, likely because the watermark is embedded as a targeted misclassification specific to one classifier's energy landscape and does not transfer across misaligned landscapes. This indicates that only the defender's private classifier can verify the watermark, preventing adversaries from forging it with a substitute classifier.

## G. Generalization Analysis

### G.1. Generality across Model Architectures

Although Cert-LAS is implemented on SD v1.4 by default to ensure fair comparison with prior watermarking baselines, it is not inherently tied to this backbone: the LFS indicator only relies on the layerwise update consistency of diffusion models under fine-tuning. To examine whether this property is specific to the SD v1.4 UNet, we evaluate Cert-LAS on Stable Diffusion XL (SDXL) (Podell et al., 2023) and Sana-1.6B (Xie et al., 2024), representative models

of the two mainstream T2I diffusion architectures (UNet-based and Transformer-based). Notably, on Sana-1.6B we instantiate the verification classifier with a flow-matching velocity-MSE score, rather than the denoising score used for SD v1.4/SDXL. We find that the same layerwise consistency holds on both backbones, and hence LFS-guided smoothing transfers across architectures.

**Layerwise Consistency.** We repeat the pilot study of Appendix C on both backbones, using the normalized stability score $S(l) = 1 - \mathrm{RD}(l)/(L-1)$ to account for their different layer counts. As Fig. 9 shows, the $S(l)$ distributions of SDXL and Sana-1.6B almost coincide with that of SD v1.4 at high stability scores, indicating that the layerwise consistency is an architectural property shared across UNet- and Transformer-based diffusion models rather than an artifact of the SD v1.4 UNet.

**Watermark Robustness.** We then evaluate watermark robustness under both standard full fine-tuning (Tab. 15) and advanced fine-tuning (Tab. 16). Across all datasets and fine-tuning paradigms, Cert-LAS remains highly robust on both backbones. Specifically, even under the most challenging regime of full fine-tuning for 2000 steps, T@$10^{-6}$F stays above 0.988 for both SDXL and Sana-1.6B, while advanced fine-tuning (DreamBooth, LoRA, Custom Diffusion) leaves it essentially unaffected. This indicates that Cert-LAS transfers effectively across architectures and scales, confirming that the proposed LFS-guided layer-adaptive smoothing is not restricted to the UNet backbone.

### G.2. Generality across Classification Tasks

Although Cert-LAS adopts a Dogs-vs-Cats class pair with a fixed prompt template by default, the class pair, prompt template, and number of classes are design choices of the

*Table 16.* Cross-architecture robustness (T@$10^{-6}$F) under advanced fine-tuning: DreamBooth, LoRA, and Custom Diffusion.

*(a) DreamBooth*

| Backbone | 250 | 500 | 750 | 1000 |
|---|---|---|---|---|
| SDXL | 1.000 | 1.000 | 1.000 | 1.000 |
| Sana-1.6B | 1.000 | 1.000 | 1.000 | 1.000 |

*(b) LoRA (rank 640)*

| Backbone | 500 | 1000 | 1500 | 2000 |
|---|---|---|---|---|
| SDXL | 1.000 | 1.000 | 1.000 | 1.000 |
| Sana-1.6B | 1.000 | 1.000 | 1.000 | 0.998 |

*(c) Custom Diffusion*

| Backbone | 100 | 200 | 300 | 400 | 500 |
|---|---|---|---|---|---|
| SDXL | 1.000 | 1.000 | 1.000 | 1.000 | 1.000 |
| Sana-1.6B | 1.000 | 1.000 | 1.000 | 1.000 | 1.000 |

*Table 17.* Generality of Cert-LAS across classification tasks, prompt templates, and the number of classes. Template 1 = "{class}"; Template 2 = "a blurry photo of a {class}"; Template 3 = "a photo of a {class}, a type of pet".

| Variation | Configuration | T@$10^{-6}$F↑ | VSR↑ |
|---|---|---|---|
| Class pair | Dogs-vs-Cats | 1.000 | 1.000 |
| | Black-vs-White | 1.000 | 1.000 |
| | Ship-vs-Truck | 1.000 | 1.000 |
| Prompt template | Template 1 | 1.000 | 1.000 |
| | Template 2 | 1.000 | 1.000 |
| | Template 3 | 1.000 | 1.000 |
| Multi-class | STL-10 (10-class) | 1.000 | 1.000 |
| | CIFAR-10 (10-class) | 1.000 | 1.000 |

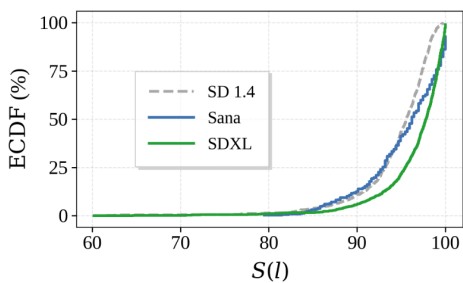

*Figure 9.* Empirical cumulative distribution functions of the stability score $S(l)$ across layers for SDXL and Sana-1.6B.

defender rather than inherent constraints of the method. To verify this, we jointly embed Cert-LAS on the same backbone under three variations: (i) multiple alternative binary class pairs, (ii) varied prompt templates, and (iii) multi-class settings on STL-10 and CIFAR-10. As shown in Tab. 17, all configurations achieve T@$10^{-6}$F = 1.000 and VSR = 1.000. This indicates that Cert-LAS generalizes across diverse class pairs, prompt templates, and multi-class settings, and is not tied to the default Dogs-vs-Cats instance.

## H. Extended Discussion on Threat Model

Arguably, the MOV setting adopted in the main paper is compatible with realistic legal and compliance-oriented forensics: a trusted verification authority can obtain an executable copy of the suspect model (*e.g.*, a weight snapshot or an equivalent implementation) via forensic procedures, platform retention, or lawful requests, and perform verifiable ownership determination without disclosing the owner's private verification information. In particular, when direct access to the model's internal parameters is unavailable and verification can only proceed via querying the suspect model, our method could degrade to an *empirical* watermarking approach. The owner queries the suspect model using class prompts, collects the generated outputs, and applies a private diffusion classifier for decision-making. Although this variant no longer provides certified robustness guarantees without inference-time layer-adaptive smoothing, it remains highly effective in practice: as demonstrated in our experiments (in Section 5.3), the degraded method exhibits strong empirical robustness against various fine-tuning attacks, while the trigger-free design ensures stealthiness by employing neither explicit triggers on the input side nor visible artifacts in generated images, thereby evading input/output-space watermark auditing (in Section 5.2).

## I. Extended Discussion: Multi-Owner Scenarios

Our Cert-LAS mainly targets model ownership verification, *i.e.*, determining whether a suspect model derives from a protected one, rather than owner identification among a large pool. Nevertheless, the multi-owner scenario, where distinct owners' watermarks must remain mutually distinguishable, is also worth exploring. Cert-LAS achieves such non-interference along two orthogonal dimensions: the task dimension and the classifier dimension. Along the task dimension, the generality across class pairs (Tab. 17) implies that different owners can adopt non-overlapping task configurations, keeping their watermark responses separable without degrading verifiability. Along the classifier dimension, as established in Appendix F.2 (Tab. 14), the verification signal is uniquely bound to the watermarking-stage private classifier, and replacing it at inference renders the watermark undetectable. This non-transferability ensures that distinct owners' watermarks do not interfere with one another. Nevertheless, how to enable scalable multi-owner verification remains an important open problem.

## J. Qualitative Results for Downstream Fine-tuning

In this section, we provide qualitative evidence that our watermark embedding does not degrade the model's adaptability for downstream customization. Fig. 10 shows gen-

*(a)* Advanced Fine-tuning via LoRA (Rank = 320)

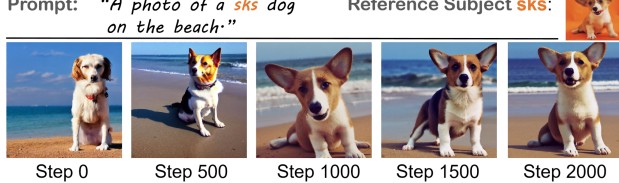

*(b)* Advanced Fine-tuning via DreamBooth

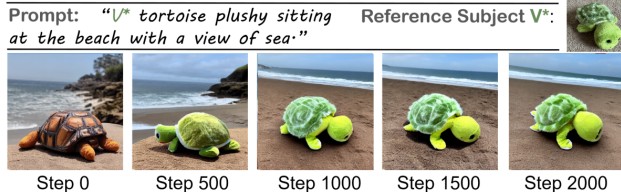

*(c)* Advanced Fine-tuning via Custom Diffusion

*Figure 10.* Qualitative generations of SD v1.4 watermarked by Cert-LAS after downstream fine-tuning with (a) LoRA, (b) Dream-Booth, and (c) Custom Diffusion. The embedded watermark does not impair the model's adaptability for downstream customization.

erations from SD v1.4 watermarked by Cert-LAS, subsequently fine-tuned using three popular personalization techniques: LoRA, DreamBooth, and Custom Diffusion. Across all three fine-tuning paradigms, the watermarked model successfully learns the target concepts and produces high-quality, style-consistent outputs, demonstrating that Cert-LAS imposes no observable degradation.

