# OpenReview forum: "Cert-LAS: Toward Certified Model Ownership Verification for Text-to-Image Diffusion Models via Layer-Adaptive Smoothing"
_ICML.cc/2026/Conference — ICML 2026 regular_

### Official Review · Reviewer_jvc3 · 2026-03-09

**Soundness:** 3
**Presentation:** 3
**Significance:** 3
**Originality:** 3
**Overall Recommendation:** 4
**Confidence:** 4

**Summary:**

This paper proposes Cert-LAS, a certified model ownership verification method for T2I diffusion models. This paper borrows the idea of certified robustness from image classifiers and apply it to generative diffusion models to address the key limitations in existing backdoor works. Specifically, Cert-LAS uses a frozen diffusion model as a private classifier for trigger-free watermark embedding, and introduces layer-adaptive smoothing guided by a Layer Fine-tuning Sensitivity (LFS) indicator to enlarge the certifiable region. Theoretical analysis provides a closed-form verification threshold and certified radius with tightness guarantee. Experiments on SD v1.4 demonstrate strong effectiveness, stealthiness, and robustness against various attacks.

**Compliance With Llm Reviewing Policy:**

Affirmed.

**Final Justification:**

I thank the authors for the detailed rebuttal. Most of my concerns have been well addressed. For W1/Q1, the extensions via task and classifier dimensions are interesting and show promise, though they remain preliminary. This is acceptable as a limitation of the current work. As a minor suggestion for future work, it would be helpful to measure the actual Mahalanobis norm of parameter displacements under typical fine-tuning settings and report how they compare to R̄, which could further strengthen the practical interpretability of the certified radius.

Overall the paper is solid and I have raised my score to 4 (Weak accept).

**Key Questions For Authors:**

**Q1:** How can Cert-LAS be extended to support multi-bit watermarking? With only binary classification, how do you handle the scenario where multiple owners need to verify different models?

**Q2:** In Table 7, Cert-LAS (w/o smooth) drops to 0.000 even at L2 = 0.2, but Cert-LAS (w/ smooth) maintains 1.000. Does this mean the watermark signal itself is actually very fragile, and all the robustness comes from the smoothing at inference time?

**Q3:** What is the computational cost of the verification phase? With M = 100 noise samples and N = 1000 images, each requiring full diffusion generation and classification, the total cost seems very high.

**Limitations:**

yes

**Strengths And Weaknesses:**

**Strengths:**

**S1:** The paper clearly demonstrates the limitations of existing methods through Section 3. The experiments on trigger detectability (Table 1) and watermark fragility (Figure 1) are convincing and provide strong motivation.

**S2:** Using the diffusion model itself as a classifier to define the watermark signal is an elegant idea. This avoids the need for synthetic triggers and naturally achieves low suspiciousness scores.

**S3:** The paper provides a complete theoretical pipeline: from the closed-form verification threshold (Theorem 4.8) to the certified radius (Theorem 4.9) to the tightness proof (Theorem A.8). The proofs in the appendix are detailed. The connection between LFS-guided noise allocation and the enlargement of the certifiable region is well-reasoned.

**S4:** The evaluation covers many practical scenarios including four fine-tuning datasets, three advanced fine-tuning methods (LoRA, DreamBooth, Custom Diffusion), and intentional attacks (PGD, adversarial directions).

**S5:** The exponential growth schedule reduces training time, which makes the method practical. The observation about flat loss basins of pretrained models is insightful and well-utilized.

**Weaknesses:**

**W1:** The entire method relies on a binary prompt set Y = {"cat", "dog"}. This is a very restrictive setting. In practice, if every model owner can only use one of two classes, the watermark carries essentially 1 bit of information. How can different owners distinguish their models from each other? The paper does not discuss how to extend to multi-class or multi-bit watermarking. This is a fundamental limitation for real-world deployment.

**W2:** The paper reports $\bar{R}$ values like 1.48 or 1.68, but it is hard for readers to understand what these numbers mean in practice. For example, what magnitude of real-world fine-tuning corresponds to $\bar{R} = 1.48$ in the Mahalanobis norm? The paper would benefit from a concrete mapping between the certified radius and actual fine-tuning budgets (e.g., how many steps of LoRA fine-tuning falls within the certified region).

**W3:** While there is no synthetic trigger token in the prompt, the verifier still needs to query the model with specific class prompts ("a photo of a cat"). In some sense, the class prompt itself serves as a trigger. The distinction from traditional trigger-based methods should be discussed more carefully.

**W4:** Additionally, if an adversary knows the method uses diffusion classification with common class prompts, they could potentially audit the model's classification behavior.

**W5:** There are some typos in the paper:

1) On Line 52, Page 1, "intentional or unintentional" should be "intentionally or unintentionally".

2) On Line 100, page 2, "We prove" should be "we prove".

3) On Line 298, Page 6, in the sentence "Given an evaluation class prompt $y \in \mathcal{Y}$, let ..." the outer variable $y$ (denoting the evaluation class prompt) and the optimization variable $y$ inside the $\arg\max$ share the same symbol, which creates a notational conflict.

---

> ### Author Rebuttal · Authors · 2026-03-31
>
> Thank you for your thorough review. We address your concerns as follows.
>
> **Response to W1&Q1: Multi-Bit Watermarking and Multi-Owner Scenarios**
>
> Thank you for this important question!
>
> - **Scope clarification.** Cert-LAS targets model ownership verification (whether a suspect model derives from a protected one), not owner identification among a large pool.
> - That said, multi-owner scenarios are important. **Cert-LAS admits two natural extensions**:
>   - **Task dimension.** We jointly embedded multiple classification pairs and multi-class settings on the same model. **VSR reach 1.000 across all settings** (see Reviewer jbqZ, W2).
>   - **Classifier dimension.** Replacing the private classifier at inference yields TPR near 0 (see Reviewer jbqZ, W4). Since different classifiers' latent spaces are not aligned, the watermark is invisible to other classifiers. **This non-transferability naturally guarantees non-interference across owners.**
>
> - **Limitation.** Simply increasing prompt pairs or classifiers does not scale ideally for large-scale multi-bit authorization. A shared-backbone framework with lightweight user-specific private information is a promising direction for future work.
>
> ---
>
> **Response to W2: Practical Interpretation of Certified Radius**
>
> Thank you for this question! The certified radius R̄ quantifies parameter displacement under the layer-adaptive Mahalanobis norm, characterizing the magnitude of parameter modification rather than the number of fine-tuning steps. Even with identical step counts, different update directions, optimization paths, and datasets can yield substantially different displacements. **Consequently, R̄ does not admit a direct mapping to a fixed number of fine-tuning steps.**
>
> ---
>
> **Response to W3: Distinction from Traditional Trigger-Based Methods**
>
> Thank you for pointing this out!
>
> - **Clarification.** As discussed in Appendix D.2, existing triggers include synthetic triggers (e.g., rare tokens) and concept triggers. Our "trigger-free" claim refers to being free of synthetic triggers.
> - **Distinction.** The class prompts can be viewed as concept triggers. However, they are naturally occurring semantic categories that do not exhibit the semantic atypicality detectable through input-space auditing. Moreover, stably training a verifiable watermark using concept triggers is itself non-trivial, since pretrained concepts are generally difficult to learn as trigger conditions.
>
> ---
>
> **Response to W4: Adversarial Auditing of Classification Behavior**
>
> Thank you for raising this point!
>
> - **Privacy of the classifier.** Even if an attacker knows the general form of diffusion classification, effective auditing remains infeasible because the verification signal is bound to the private classifier φ's latent/energy landscape, which cannot be reproduced with any public classifier. As shown in Reviewer jbqZ W4, **T@10⁻⁶F drops to nearly 0 across all classifier replacements**.
> - **Stronger threat model.** If the attacker obtains both parameters and φ, the goal shifts to removal. Our certified guarantee ensures breaking verification requires perturbations exceeding the certified radius. As shown in Fig. 3 and Tab. 7, Cert-LAS (w/ smooth) maintains T@10⁻⁶F = 0.965 at ℓ₂ budget = 0.8.
>
> ---
>
> **Response to W5: Typos and Notational Issues**
>
> Thank you for the careful proofreading! All typos and the notational conflict will be corrected in the revision.
>
> ---
>
> **Response to Q2: Source of Robustness**
>
> Thank you for this insightful question!
>
> - **Both phases contribute.** "Cert-LAS (w/o smooth)" in Tab. 7 only removes inference-time smoothing; the model is still robustly trained. As shown in Tab. 3–6, w/o smooth remains stable under standard fine-tuning. We further trained a model without any smoothing, and verification drops to 0 at 2000 steps, confirming **training-phase smoothing injects substantial empirical robustness**.
>
> | Dataset | 200 | 500 | 1000 | 1500 | 2000 |
> |---|---|---|---|---|---|
> | CelebA-HQ | 0.459 | 0.301 | 0.182 | 0.078 | 0.000 |
> | Landscape | 0.748 | 0.593 | 0.252 | 0.144 | 0.000 |
>
> - **Role of inference-time smoothing.** Tab. 7 considers worst-case adversarial perturbations where empirical robustness alone is insufficient. Inference-time smoothing aggregates noisy predictions via majority voting, elevating robustness into a **provable guarantee within the certified radius**.
>
> ---
>
> **Response to Q3: Computational Cost of Verification**
>
> Thank you for this reminder! The default (M=100, N=1000) adopts a unified protocol for baseline comparison, not a practical requirement.
>
> - **Diminishing returns.** As shown in Tab. 10, (M=100, N=100) reduces time from 21.10h to 2.22h while VSR remains nearly unchanged.
> - **Comparable overhead.** 2.22h is on the same order as existing methods (e.g., [2]: ~1.43h). We will explore further efficiency optimization in future work.
>
> [2] Wang et al. "SleeperMark: Towards Robust Watermark against Fine-Tuning T2I Diffusion Models." CVPR, 2025.

---

> > ### Author Rebuttal · Reviewer_jvc3 · 2026-04-04
> >
> > I thank the authors for their detailed responses. Several of my concerns have been addressed.
> >
> > My concern regarding W2 remains unresolved. The authors' response essentially states that the certified radius cannot be mapped to fine-tuning steps, which does not address my original question about the practical interpretability of R.
> >
> > Regarding W1/Q1, I appreciate the honesty in acknowledging the limitation. The extensions via task and classifier dimensions are interesting but remain preliminary. This is acceptable as a limitation of the current work, though it does constrain the practical applicability of the method.
> >
> > Therefore, I will raise my score to 4.

---

> > > ### Author Response · Authors · 2026-04-07
> > >
> > > Thank you for the prompt response and the detailed follow-up comments! We are glad that our rebuttal has addressed several of your concerns and we deeply appreciate your willingness to increasing your score. We are more than happy to further address the remaining concerns as follows.
> > >
> > > ---
> > >
> > > **Q1:** My concern regarding W2 remains unresolved. The authors' response essentially states that the certified radius cannot be mapped to fine-tuning steps, which does not address my original question about the practical interpretability of $\bar{R}$.
> > >
> > > **R1:** Thank you for your follow-up and we do understand your concerns!
> > >
> > > - **As a first step toward certified watermark for T2I diffusion model, we have to admit that establishing a one-to-one mapping between $\bar{R}$ and fine-tuning steps is difficult.** The certified region is a Mahalanobis ellipsoid in parameter space, while fine-tuning trajectories depend on the model initialization and the downstream dataset, leading to diverse optimization paths. Some paths may orbit around the ellipsoid without exiting it even after many steps, making any one-to-one mapping between $\bar{R}$ and fine-tuning steps inherently unreliable.
> > > - Despite this, **$\bar{R}$ retains practical interpretability: it defines the maximum perturbation range in parameter space within which ownership verification is guaranteed**. Specifically, as long as the total parameter change remains within this bound, our method provably ensures correct ownership verification, even under the worst-case threat model where the attacker has full knowledge.
> > > - **Arguably, $\bar{R}$ remains heuristically meaningful.** As a pioneering attempt to bring certified robustness to T2I diffusion model watermarking, our work highlights the vulnerability of existing empirical methods and takes an initial step toward certified protection of generative model IP. In this sense, $\bar{R}$ is meaningful because it provides a principled robustness quantity for a problem that previously lacked certified guarantees. Nevertheless, we fully agree that establishing a mapping between $\bar{R}$ and fine-tuning steps is a valuable direction for future work.
> > >
> > > ---
> > >
> > > **Q2:** Regarding W1/Q1, I appreciate the honesty in acknowledging the limitation. The extensions via task and classifier dimensions are interesting but remain preliminary. This is acceptable as a limitation of the current work, though it does constrain the practical applicability of the method.
> > >
> > > **R2:** Thank you for recognizing the value of our preliminary experiments! We are glad that the extensions via task and classifier dimensions have initially demonstrated the feasibility of this direction. We will include a more thorough discussion on its current limitations and potential solutions in the revision, and pursue more scalable multi-owner watermarking in future work.

---

### Official Review · Reviewer_jbqZ · 2026-03-20

**Soundness:** 2
**Presentation:** 3
**Significance:** 3
**Originality:** 3
**Overall Recommendation:** 4
**Confidence:** 3

**Summary:**

This paper introduces model ownership verification for text-to-image diffusion models under post-release parameter perturbations. It argues that prior backdoor-style diffusion watermarking often relies on a faithful verification assumption and can fail when the suspect model is modified. It proposes Cert-LAS, a trigger-free model ownership verification framework with two stages.
1. watermark embedding using a private diffusion classifier and layer-adaptive smoothing guided by layer fine-tuning sensitivity.
2. ownership verification by comparing the watermark robustness of a suspect model against the reference probability of an unwatermarked reference under the same smoothing, followed by a paired statistical test.

The paper also provides a closed-form verification threshold and a certified-radius style analysis under Gaussian layer-adaptive perturbations. Experiments on SD v1.4 report improved robustness against downstream fine-tuning and parameter perturbation attacks compared to existing baselines.

**Compliance With Llm Reviewing Policy:**

Affirmed.

**Final Justification:**

After considering both the paper and the authors’ rebuttal, I am raising my overall evaluation to 4. My main concern in the original review was whether the WR–RP separation is truly attribution-specific, rather than partly explained by model mismatch, distribution shift, or coupling between the private classifier and the reference generator. The rebuttal does not completely eliminate this concern, but it does address it more substantially than before. In particular, the additional evidence on broader non-watermarked suspects, the clarification that (\tau) is determined by the testing setup rather than tuned manually, and the expanded discussion of decoupled classifier/reference-generator variations make the empirical case materially stronger. The clarification on the role of (q^*) is also helpful, especially in distinguishing the certified WR/RP/VSR pipeline from the confidence-based (T@10^{-6}F) metric used for comparison with prior empirical baselines.  So my final position is more positive than before, but still somewhat cautious on soundness at the attribution level. Overall, however, I believe the rebuttal has addressed enough of my core concerns to improve my assessment: the paper is original, practically relevant, and promising, and the remaining weaknesses seem more like limitations to clarify and expand on in revision than fatal flaws.

**Key Questions For Authors:**

1. How is $q^*$ chosen (e.g., why $[0.45,0.55]$), and how sensitive are WR/RP separation and final decisions to this choice?
2. What is the false-positive rate on independently trained non-watermarked suspects under distribution shift, including cross-architecture, cross-dataset, and different pretraining lineages (even with similar visual outputs)?
3. Can you ablate the effect of using the same frozen model as both the private classifier and the reference generator versus decoupled models? This would clarify whether WR-RP gains are attribution-specific or partly due to coupling bias.

**Limitations:**

While the paper evaluates robustness under multiple perturbation settings, the limitation discussion should more explicitly address the following aspects:
1. attribution reliability under distribution/model shift (e.g., independently trained non-watermarked suspects),
2. potential bias from using the same frozen model as both the private classifier and the reference generator,
3. sensitivity to key design choices such as $q^*$ and prompt construction.

A clearer calibration protocol and false-positive risk analysis are required for high-stakes ownership claims.

**Strengths And Weaknesses:**

### Strengths
1. The paper addresses a realistic threat model (post-release modifications) and moves beyond assumptions of purely benign fine-tuning.
2. The method integrates statistical testing, robustness-aware training, and certification analysis into a coherent pipeline.

### Weaknesses
1. A central attribution assumption is under-validated. The paper does not convincingly show that WR-RP separation is attribution-specific, rather than being driven by distribution or model mismatch.
2. The binary target construction appears highly task-specific (Dogs-vs-Cats, fixed prompts, fixed $q^*$ ), but the sensitivity to $q^*$ and prompt-set design is not thoroughly characterized.
3. The paper lacks sufficient false-positive evaluation on independently trained, non-watermarked suspect models under cross-architecture and cross-data shifts.
4. The same frozen model appears to serve as both the private classifier and the reference generator, which may introduce coupling bias in WR-RP attribution and deserves an explicit ablation.

---

> ### Author Rebuttal · Authors · 2026-03-31
>
> Thank you for your thorough review. We address your concerns as follows.
>
> **Response to W1: Attribution Mechanism and WR-RP Separation**
>
> Thank you for this insightful question!
>
> - **Attribution mechanism.** The WR-RP separation stems from the stability of diffusion classifier predictions. As [1] demonstrate, diffusion classifiers classify by modeling the data distribution, making their responses inherently robust to model mismatch.
> - **Cross-architecture validation.** We replaced the reference generator with different architectures and scales. WR remains at 1.000 while RP stays below 0.15, **confirming the gap is attribution-specific, not an artifact of model mismatch**.
>
> | Ref. Generator | WR | RP |
> |---|---|---|
> | SD v1.4 | 1.000 | 0.125 |
> | SDXL | 1.000 | 0.109 |
> | Z-Image | 1.000 | 0.144 |
>
> [1] Chen et al. "Robust Classification via a Single Diffusion Model." arXiv, 2023.
>
> **Response to W2: Generality of Target Construction and Prompt Design**
>
> Thank you for pointing this out!
>
> - **Not task-specific.** The target construction is privately designed by the defender. Dogs-vs-Cats is simply the demonstration instance, not a limitation.
> - **Alternative class pairs.** We tested alternative binary tasks, **all achieving VSR = 1.000**, confirming no dependence on any particular class pair.
> - **Prompt template stability.** Alternative templates also consistently achieve 1.000.
> - **Multi-class extension.** STL-10 and CIFAR-10 also achieve VSR = 1.000, confirming extension to multi-class scenarios.
>
> | Classification Task | T@10⁻⁶F | VSR |
> |---|---|---|
> | Dogs-vs-Cats / Black-vs-White / Ship-vs-Truck | 1.000 | 1.000 |
> | STL-10 (10-class) / CIFAR-10 (10-class) | 1.000 | 1.000 |
>
> | Prompt Template | T@10⁻⁶F | VSR |
> |---|---|---|
> | {class} / a blurry photo of a {class} / a photo of a {class}, a type of pet | 1.000 | 1.000 |
>
> **Response to Q1: Choice and Sensitivity of q\***
>
> TThank you for this insightful question!
>
> - **Design rationale.** Unwatermarked model confidences are nearly uniform (~0.500), so a mild target stably above 0.5 suffices.
> - **Role of q\*.** q\* only controls watermark signal strength during training. Verification uses predicted class labels only, so q\* does not participate in the verification decision.
> - **Sensitivity analysis.** Higher q\* leads to faster VSR saturation but increases DreamSim, since classification loss increasingly dominates perceptual regularization. **q\* = 0.55 balances verification robustness and fidelity.**
>
> | q\* | VSR | DreamSim |
> |---|---|---|
> | 0.51 | 0.352 | 0.075 |
> | 0.53 | 0.833 | 0.096 |
> | 0.55 | 1.000 | 0.111 |
> | 0.70 | 1.000 | 0.138 |
> | 0.85 | 1.000 | 0.149 |
> | 1.0 | 1.000 | 0.154 |
>
> | cat confidence | dog confidence |
> |---|---|
> | 0.50000945 | 0.49999055 |
> | 0.50001873 | 0.49998127 |
> | 0.50002503 | 0.49997497 |
>
> **Response to W3&Q2: False-Positive Evaluation**
>
> Thank you for pointing this out!
>
> - **Cross-architecture false positives.** We tested unwatermarked models as suspects. **None exceed the threshold τ from Theorem 4.8**, indicating low false-positive risk.
>
> | Suspect Model | WR | RP | τ |
> |---|---|---|---|
> | SD1.4 | 0.106 | 0.125 | 0.426 |
> | SDXL | 0.213 | 0.125 | 0.426 |
> | Z-Image | 0.008 | 0.125 | 0.426 |
>
> - **Private-classifier specificity.** Replacing the private classifier at inference with different sources, **WR does not exceed τ even with the same method but a different classifier**. These results confirm Cert-LAS neither misidentifies unwatermarked models nor falsely attributes other watermarked models.
>
> | Cls. (replaced) | WR | RP | τ |
> |---|---|---|---|
> | SD1.4 (v1) | 0.004 | 0.102 | 0.351 |
> | SD1.4 (v2) | 0.048 | 0.008 | 0.219 |
> | SD1.4 (v3) | 0.019 | 0.197 | 0.456 |
> | SDXL | 0.232 | 0.210 | 0.469 |
> | Z-Image | 0.001 | 0.003 | 0.233 |
>
> **Response to W4&Q3: Coupling Bias Ablation**
>
> Thank you for your question and we do understand your concerns!
>
> - **Replacing reference generator.** As shown in W1, substituting with SDXL/Z-Image keeps WR=1.000 and RP<0.15, indicating the **WR-RP gap does not depend on the reference generator**.
> - **Replacing training-time classifier.** Re-embedding with SDXL/Z-Image maintains WR=1.000 with clear separation, confirming **effective embedding does not rely on a single model serving both roles**.
>
> | Cls. (training) | WR | RP |
> |---|---|---|
> | SD 1.4 | 1.000 | 0.125 |
> | SDXL | 1.000 | 0.210 |
> | Z-Image | 1.000 | 0.013 |
>
> - **Replacing inference-time classifier.** As shown in W3&Q2, WR drops to near zero, demonstrating **strong private-classifier specificity**.

---

> > ### Author Rebuttal · Reviewer_jbqZ · 2026-04-04
> >
> > The rebuttal is helpful in several respects. The added \(q^*\) sensitivity analysis, alternative class/prompt experiments, and multi-class results reduce my concern that the method only works in a narrow setting. The additional false-positive checks on unwatermarked suspects also improve the empirical calibration story.
> >
> > However, my main concern about attribution specificity is only partially resolved.
> > - The new cross-model WR/RP results are encouraging, but they still do not fully show that the WR-RP gap is ownership-specific rather than partly caused by model/data mismatch under distribution shift.
> > - The false-positive study is still limited. I would find it more convincing to test independently trained, non-watermarked suspects across broader architectures, datasets, and pretraining lineages.
> > - The coupling-bias issue also remains only partially addressed. The replacement experiments are useful, but I still do not see a clean decoupled ablation that independently varies the private classifier and the reference generator while keeping the rest of the pipeline fixed.
> > - The clarification on \(q^*\) is helpful, but the paper should more clearly distinguish the label-based WR/RP/VSR verification rule from the confidence-based \(T@10^{-6}F\) evaluation.
> >
> > ### Follow-up questions for the authors
> > 1. Can you provide a direct decoupled ablation where the private classifier and reference generator are independently varied while the rest of the pipeline is fixed?
> > 2. Can you expand the false-positive evaluation to independently trained, non-watermarked suspects with different architectures, datasets, and pretraining lineages?
> > 3. Is the threshold \(\tau\) recalibrated for each classifier/reference pairing? If so, what is the calibration protocol and how stable is it?
> > 4. Can you clarify whether and how \(q^*\) affects the confidence-based \(T@10^{-6}F\) evaluation?
> >
> > Overall, the rebuttal improves the empirical picture, so I will raise my score to 4.

---

> > > ### Author Response · Authors · 2026-04-07
> > >
> > > Thank you for the detailed follow-up! We are glad that our rebuttal has addressed some of your concerns, and deeply appreciate your openness to raising the score. We are more than happy to further address the remaining concerns below.
> > >
> > > ---
> > > **Q1:** Isolating ownership-specific signal from distribution-shift in WR–RP gap.
> > >
> > > **R1:** Thank you for the insightful comment! While proving the gap is entirely free from distribution-shift is hard given generative model complexity, the following evidence strongly supports it is mostly ownership-specific:
> > > - **The gap is mainly attributed to ownership-specific watermarking.** By design, only an ownership-specifically watermarked generator can elevate WR, while unwatermarked ones are correctly classified by default (W3&Q2), as diffusion classifiers are zero-shot [1][2].
> > > - **To further alleviate your concern, we conduct additional ablations:**
> > >     - **Varying the model.** Beyond SDXL and Z-Image (W3&Q2), we fine-tune SD v1.3–v1.5 on 3 datasets into 9 architecturally similar (hence hardest-to-distinguish) reference generators. Gaps stay within 0.864–0.899, **indicating negligible model-mismatch effect**.
> > >     - **Varying the data.** Since the class pair governs generated data distribution, we vary it on the same SD v1.4 (W2); gaps remain consistent, **indicating negligible data-mismatch effect**.
> > >
> > > [1] Li et al. ICCV 2023; [2] Clark & Jaini. NeurIPS 2024.
> > >
> > > ---
> > > **Q2:** False positive evaluation across broader suspects.
> > >
> > > **R2:** Thank you for the insightful question! We fully understand that FPR is critical in practice.
> > > - **We respectfully note that our evaluation already covered mainstream T2I architectures.** We evaluated SOTA models from UNet (SDXL) and Transformer (Z-Image), independently pretrained with distinct data and lineages from SD v1.4. Their WR values stay below τ = 0.426 (W3&Q2), indicating no false attribution.
> > > - Arguably, **false positives are theoretically controlled, not merely empirically observed.** Via a paired-sample t-test, Theorem 4.8 derives a closed-form τ at significance level α bounding the FPR.
> > > - **To further alleviate your concern, we expand the evaluation to broader suspects.** Specifically, we cover 5 independently pretrained models (UNet: SD v1.3–v1.5; Transformer: Z-Image, Sana), each fine-tuned on 3 datasets, yielding 15 non-watermarked suspects (τ = 0.426). **All WR values remain far below τ (0.005–0.127), further indicating no false attribution across broader suspects.**
> > > ---
> > > **Q3:** Clean Decoupled ablation.
> > >
> > > **R3:** Thank you for pointing it out and we apologize for the prior unclear response. Each ablation is already cleanly decoupled: only the specified component varies, with the rest at default.
> > > - **Private classifier:** The WR–RP gap remains above 0.79 when varying the embedding-time classifier (W4&Q3), **decoupling from the embedding-time classifier**. In contrast, varying the inference-time classifier across 3 finetuned SD v1.4, SDXL, and Z-Image collapses the gap in all cases (W3&Q2), indicating it is **uniquely tied to the private classifier and cannot be reproduced by others.**
> > > - **Reference generator(W1, W4&Q3):** The gap remains above 0.85 when varying the reference generator with SDXL and Z-Image, **decoupling from the reference generator.**
> > > ---
> > > **Q4:** Label-based WR/RP/VSR vs. confidence-based T@10⁻⁶F.
> > >
> > > **R4:** Thank you for this constructive suggestion! We apologize for any prior misunderstanding and clarify below:
> > > - **In practice, the model owner can directly access full prediction logits of the private classifier**, making confidence-based T@10⁻⁶F evaluation feasible.
> > > - **WR/RP/VSR and T@10⁻⁶F target certified and empirical methods, respectively.** Since existing baselines [3,4] are empirical, we report T@10⁻⁶F for fair comparison. Cert-LAS is certified, so we additionally report WR/RP/VSR for certified robustness. We'll clarify this in the revision.
> > >
> > > [3] Zhao et al. arXiv 2023; [4] Wang et al. CVPR 2025.
> > >
> > > ---
> > > **Q5:** Is τ empirically recalibrated?
> > >
> > > **R5:** Thank you for this insightful question and we apologize for any prior ambiguity. **τ is not a hyperparameter but is automatically determined by the hypothesis testing setup.** As shown in Theorem 4.8, τ is computed in closed form from the significance level, the upper bound on RP, and the sampling budget.
> > >
> > > ---
> > > **Q6:** Effect of q* on confidence-based T@10⁻⁶F.
> > >
> > > **R6:** Thank you for this insightful question!
> > > - **q\*  controls the watermark strength during training and does not enter verification directly.** A larger q* lets the classification loss dominate the perceptual loss, yielding higher-confidence misclassification and higher T@10⁻⁶F.
> > > - **To further alleviate your concern, we additionally report T@10⁻⁶F under the same q\*ablation as in Q1.** As q* increases (0.51 → 0.53 → 0.55 → 0.70 → 1.0), T@10⁻⁶F rises monotonically (0.504 → 0.977 → 1.000 → 1.000 → 1.000), near-saturating already at 0.53, indicating **T@10⁻⁶F is insensitive to q\* once above a small threshold.**

---

### Official Review · Reviewer_8zbA · 2026-03-22

**Soundness:** 2
**Presentation:** 3
**Significance:** 3
**Originality:** 4
**Overall Recommendation:** 4
**Confidence:** 4

**Summary:**

First, the authors study existing model ownership verification methods for text-to-image diffusion models and demonstrate their limitations. In response, the authors propose Cert-LAS, which is the first certified watermarking method for model ownership verification designed for text-to-image diffusion models. Cert-LAS relies on a diffusion classifier as an implicit watermark carrier and employs layer-adaptive randomized smoothing. This smoothing mechanism is specifically tailored to UNet architectures. Additionally, the authors provide robustness guarantees for Cert-LAS that hold under certain well-analysed conditions. Finally, the authors conduct extensive experiments to validate the effectiveness of Cert-LAS on SD1.4.

**Compliance With Llm Reviewing Policy:**

Affirmed.

**Final Justification:**

I feel that the paper would be much stronger if the evaluation on modern models was integrated as a core part of this contribution. This is, of course, impossible in the limited time of the rebuttal. That said, the provided evidence that the method generalizes to SOTA models is compelling. My concerns have been adequately addressed in the rebuttal. I still view the paper as a borderline case, especially given the extensive work needed to fully incorporate the content of the rebuttal in the updated version. However, I now lean towards accepting the paper at ICML, which is reflected by my Weak Accept final score.

**Key Questions For Authors:**

Q1: Can the authors provide any empirical (preferred) or theoretical evidence for the effectiveness of their method for **current SOTA t2i diffusion models?** SD1.4 Can no longer be considered SOTA.

Q2: Can the authors provide any empirical (preferred) or theoretical evidence for the effectiveness of their method for t2i diffusion models **beyond U-Net based architecures** (e.g., transformer-based)? Most SOTA models are currently based on the transformer architecture.

Q1 and Q2 are tightly connected as current SOTA diffusion models are, in general, transformer-based. Thus the questions can be answered together (unless Cert-LAS is not applicable beyond U-Net by design).

Please consider:
- Stable Diffusion XL or SD2.X  if Cert-Las is not applicable beyond U-Net architectures.
- Z-Image Team, "Z-Image: An Efficient Image Generation Foundation Model with Single-Stream Diffusion Transformer". Does not have Dreambooth support, but it is usable in other experiments. Uses transformer backend.
- FLUX by BlackForest. Has Dreambooth support. Uses transformer backend.
- any other model

**I do not expect the authors to redo all the experiments for all models above.** Rather, I kindly ask to consider how to make the strongest argument possible that Cert-LAS works for the current SOTA t2i diffusion models.

To clarify: demonstrating limitations of existing model ownership verification methods on one model only (even not SOTA) was perfectly fine. I am also aware that using SD1.4 allows for readily available tools for model adaptation, such as DreamBooth.

**Limitations:**

Yes, but does not mention the point I raised as the key weakness.

**Strengths And Weaknesses:**

Strengths:
- Good paper structure that goes from identifying and demonstrating a problem with existing methods to proposing and validating a solution.
- Demonstrating the limitations of undetectability and robustness of existing methods is a valuable contribution.
- Cert-LAS provides robustness guarantees under certain-conditions that have been well analysed in the paper.
- The evaluation within the scope of the only evaluated model is extensive and well-thought-out.

Weakness:
- The Method is hard to follow and would benefit from a diagram and an algorithm block.
- The method is tailored to U-Net-based architectures, which are no longer the basis of SOTA t2i diffusion models. "[Cert-LAS] introduces layer-adaptive randomized smoothing tailored to UNet architectures." This raises the question if Cert-LAS is applicable to SOTA t2i diffusion models. (Please see Questions)
- Evaluation is only done on one model - SD1.4. SD1.4 can not currently be considered SOTA. In fact, it is also not the most capable U-Net-based model. Again, this raises the question if Cert-LAS is applicable to SOTA t2i diffusion models. (Please see Questions)


This otherwise valuable and sound contribution is undermined by the evaluation being limited to one model only. This severe weakness prohibits me from recommending acceptance at this stage of the review process. Please see Questions.

---

> ### Author Rebuttal · Authors · 2026-03-31
>
> Here's the complete version with proper math formatting:
>
> ---
>
> Thank you for your thorough review and constructive comments. We greatly appreciate your valuable feedback on improving the experimental details and presentation. We address your concerns as follows.
>
> **Response to W1: Method Clarity and Presentation**
>
> Thank you for this constructive suggestion! We clarify our method below.
>
> - Cert-LAS aims to provide certifiably robust model ownership verification for T2I diffusion models. The watermark is embedded by training the generator so that a frozen diffusion classifier misclassifies its outputs into a designated target class, requiring no explicit trigger in either input or output space.
> - To ensure this watermark survives fine-tuning, we inject per-layer Gaussian noise into the generator's weights during training and average gradients over multiple perturbations. The noise is allocated adaptively via our LFS metric: sensitive layers receive more noise and stable layers less, concentrating defense where most needed. An exponential growth schedule further reduces training time by 76.6%.
> - For verification, the same per-layer noise is injected into a suspected model, and the frozen classifier evaluates its outputs to yield the watermark response rate (WR). An unwatermarked reference undergoes the same procedure, yielding the reference probability (RP). Ownership is confirmed via paired $t$-test if WR significantly exceeds RP, with the certified radius guaranteeing this gap persists under bounded perturbations.
> - Due to limited space, we will include a pipeline diagram and pseudocode in the revision.
>
> **Response to W2&W3&Q1&Q2: Generalization to SOTA and Non-UNet Architectures**
>
> Thank you for your suggestion and we do understand your concerns!
>
> - Our choice of SD v1.4 is to ensure fair comparison with baselines ([1, 2]) by following their experimental setup. However, generalizability to SOTA and non-UNet architectures is also important. To address this, we conducted additional experiments on **SDXL** [3] (larger UNet) and **Sana-1.6B** [4] (pure Transformer/DiT).
> - **Model selection.** As suggested, we attempted Z-Image (6B) but encountered numerical instability under fp16 and prohibitive memory/speed costs under bf16 with DeepSpeed within the limited time. We therefore chose Sana-1.6B as an alternative pure Transformer architecture.
> - **Layerwise consistency.** We repeated the pilot study on SDXL (1,680 layers) and Sana-1.6B (396 layers) across three datasets. Due to the large difference in layer count, we use the normalized stability score $S(l) = 1 - \text{RD}(l)/(L-1)$ for fair comparison. Both exhibit strong cross-dataset stability: SDXL achieves a mean $S(l)$ of $\sim$0.966 and Sana-1.6B $\sim$0.952, consistent with SD v1.4. **This confirms the LFS assumption is not architecture-specific.**
> - **Watermark robustness.** Even under full fine-tuning for 2,000 steps, both models maintain $\text{TPR}@\text{FPR}=10^{-6}$ above 0.988, demonstrating that **Cert-LAS effectively transfers across architectures and scales**.
>
> | ArtBench (STF) | 500   | 1000  | 1500  | 2000  |
> | -------------- | ----- | ----- | ----- | ----- |
> | SDXL           | 1.000 | 1.000 | 0.996 | 0.988 |
> | Sana-1.6B      | 1.000 | 1.000 | 0.999 | 0.993 |
>
> | CelebA-HQ (STF) | 500   | 1000  | 1500  | 2000  |
> | --------------- | ----- | ----- | ----- | ----- |
> | SDXL            | 1.000 | 1.000 | 1.000 | 0.999 |
> | Sana-1.6B       | 1.000 | 1.000 | 1.000 | 0.995 |
>
> | DreamBooth | 250   | 500   | 750   | 1000  |
> | ---------- | ----- | ----- | ----- | ----- |
> | SDXL       | 1.000 | 1.000 | 1.000 | 1.000 |
> | Sana-1.6B  | 1.000 | 1.000 | 1.000 | 1.000 |
>
> **References:** [1] Zhao et al. "A Recipe for Watermarking Diffusion Models." arXiv, 2023.
> [2] Wang et al. "SleeperMark: Towards Robust Watermark against Fine-Tuning Text-to-Image Diffusion Models." CVPR, 2025.
> [3] Podell et al. "SDXL: Improving Latent Diffusion Models for High-Resolution Image Synthesis." ICLR, 2024.
> [4] Xie et al. "SANA: Efficient High-Resolution Image Synthesis with Linear Diffusion Transformers." ICLR, 2025.

---

> > ### Author Rebuttal · Reviewer_8zbA · 2026-04-04
> >
> > I thank the authors for the provided answers. I feel that the paper would be much stronger if the evaluation on modern models was integrated as a core part of this contribution. This is, of course, impossible in the limited time of the rebuttal. I recommend that the authors greatly revise the paper with this in mind, should the work be accepted. That said, the provided evidence that the method generalizes to SOTA models is compelling.  My concerns have been adequately addressed. Thus, I increase my score.

---

> > > ### Author Response · Authors · 2026-04-06
> > >
> > > Thank you for your positive feedback and for raising the score! We are glad that our responses have addressed your concerns. We fully agree that evaluating modern models would strengthen the paper, and we have provided more details and discussions in our revision.

---

### Official Review · Reviewer_4kjn · 2026-03-24

**Soundness:** 2
**Presentation:** 3
**Significance:** 2
**Originality:** 3
**Overall Recommendation:** 4
**Confidence:** 3

**Summary:**

The authors intend to assess a major problem regarding the vulnerability of intellectual property and model ownership verification in text-to-image diffusion models. The article attempts to consider the concept of certified robustness, often regarded as a gold standard in security, and extends it to the generative modeling setting, aiming to provide provable guarantees against practical model-tampering threats such as fine-tuning and parameter perturbations. To this end, the paper proposes a framework termed Cert-LAS, which introduces a Layer-Adaptive Smoothing mechanism to allocate perturbation noise across different layers of the UNet architecture. In addition, the method leverages a private diffusion classifier to implicitly embed and verify watermarks in a trigger-free manner. Overall, the work aims to bridge the gap between empirically driven watermarking approaches and theoretically grounded security guarantees for generative models.

**Compliance With Llm Reviewing Policy:**

Affirmed.

**Final Justification:**

After careful consideration of the authors' rebuttal and the comments from other reviewers, most of my concerns have been addressed. The authors provided additional experiments in the rebuttal. My main remaining concern is that these key supplementary results currently exist only in the rebuttal and have not yet been integrated into the main text. However, the authors have explicitly committed to incorporating them in the final version. Taking into account the assessments from other reviewers, I raise my score to 4.

**Key Questions For Authors:**

Could the authors provide a comprehensive diagram illustrating both the training and verification pipelines? In particular, it would be helpful to clarify how the noise scale sigma is allocated across different UNet layers during smoothing, and how this interacts with the forward pass in practice.

In realistic deployment scenarios, a verifier may need to handle models that have undergone more substantial transformations than fine-tuning, such as pruning or quantization. How does the proposed framework perform under such modifications? Moreover, does the reliance on a private classifier and reference generator introduce additional constraints or limitations in these settings?

The paper introduces the Layer-wise Fine-tuning Sensitivity (LFS) as a key component for noise allocation. Did the authors observe consistent sensitivity patterns across different fine-tuning strategies (e.g., DreamBooth vs. LoRA)? More broadly, is the observed sensitivity primarily layer-dependent, or could it vary significantly at a finer (e.g., parameter-level) granularity?

**Limitations:**

Yes.

**Strengths And Weaknesses:**

Strengths:

The shift from heuristic defenses to provable guarantees is both timely and conceptually meaningful.

With the increasing adoption of model-as-a-service paradigms and open-source diffusion models, protecting model intellectual property against unauthorized reuse or fine-tuning (e.g., via LoRA) is an important and practical problem for the machine learning community.

The paper presents a relatively rigorous mathematical framework by extending randomized smoothing techniques to the parameter space of high-dimensional generative models. This extension is non-trivial, especially given the complexity of diffusion-based score modeling.

Experimental results on Stable Diffusion v1.4 indicate that the proposed method achieves strong robustness while maintaining competitive generation quality (e.g., in terms of FID). The reported certified radius further provides quantitative evidence of robustness.

Weaknesses:

The primary weakness of the paper lies in its presentation. From a reviewer’s perspective, the description of the proposed Layer-Adaptive Smoothing (LAS) mechanism (Section 4) is overly dense and heavily notation-driven. In particular, the transition from the general formulation of randomized smoothing to its layer-wise instantiation in UNet is difficult to follow, making it challenging to build clear algorithmic intuition.

Lack of Visualization: The paper would benefit significantly from high-level architectural diagrams or pipeline illustrations. In its current form, the interaction between the frozen reference generator, the LAS mechanism, and the private diffusion classifier is not clearly visualized, which hinders understanding of the overall workflow and reproducibility.

The proposed layer-adaptive strategy appears to be closely tied to the UNet architecture. While UNet remains widely used, recent trends are shifting toward Transformer-based diffusion backbones (e.g., DiT). It is unclear how the proposed Layer-wise Fine-tuning Sensitivity metric and noise allocation strategy would generalize to these alternative architectures.

Although the paper introduces an “Exponential Growth Schedule” to mitigate computational costs, it does not provide sufficient quantitative analysis of training or inference overhead (e.g., wall-clock time or resource consumption). Given the already high cost of training large-scale diffusion models, this may pose a practical barrier to adoption.

---

> ### Author Rebuttal · Authors · 2026-03-31
>
> Thanks for your time and effort in reviewing our work! Below we address your concerns one by one.
>
> **Response to W1&W2&Q1: LAS Mechanism, Pipeline, and Workflow**
>
> Thank you for pointing this out!
>
> - **LAS mechanism.** Randomized smoothing certifies robustness by injecting Gaussian noise into model parameters and aggregating predictions via majority vote. LAS allocates different noise scales based on layer sensitivity, concentrating protection where most needed. LFS quantifies each layer's relative update magnitude: sensitive layers receive larger noise, stable layers less, and the certified region becomes an ellipsoid adapted to layerwise structure.
> - **Embedding phase.** The generator produces images, which the **frozen diffusion classifier** evaluates to compute a misclassification loss. **LAS** wraps this by injecting layer-wise noise at each update and averaging gradients over multiple perturbations.
> - **Verification phase.** Both the suspected model and a **frozen reference generator** produce images under the same layer-adaptive noise. The **same frozen classifier** evaluates both, and ownership is confirmed via paired t-test.
> - **Noise allocation.** A short pilot fine-tuning computes each layer's LFS once. At each training step, per-layer Gaussian noise is added directly to model weights, and gradients are averaged before updating.
> - Due to limited time, we will include a pipeline diagram and pseudocode in the revision.
>
> **Response to W3: Generalization to Transformer-Based Backbones**
>
> Thank you for this important concern!
>
> - Our choice of SD v1.4 ensures fair comparison with baselines ([1, 2]). However, generalization to Transformer-based backbones is also important. We conducted additional experiments on SDXL (larger UNet) and Sana-1.6B (DiT). See Reviewer 8zbA W2&W3 for full results.
> - **Layerwise consistency.** Both SDXL and Sana-1.6B exhibit strong cross-dataset stability (mean S(l) ~0.966 and ~0.952 respectively), confirming the LFS assumption is not architecture-specific.
> - **Watermark robustness.** Even under full fine-tuning for 2000 steps, both maintain T@10⁻⁶F above 0.988, demonstrating **effective cross-architecture transfer**.
>
> **Response to W4: Training and Inference Overhead**
>
> Thank you for this concern!
>
> - **Training.** As shown in Tab. 9, Cert-LAS with Exponential Growth Schedule achieves 2.2h, **the lowest among all methods** (76.6% reduction).
> - **Inference.** Default (M=100, N=1000) is for fair baseline comparison. Reducing to (M=100, N=100) brings time to 2.22h with no VSR loss (Tab. 10), comparable to SleeperMark [2] (~1.43h).
>
> **Response to Q2: Robustness Under Quantization and Pruning**
>
> Thank you for this question!
>
> - W8A32/W8A8 and 30%/50% pruning all preserve VSR=1.000. More aggressive settings degrade VSR but simultaneously cause notable quality loss. **Neither removes the watermark without degrading quality.**
>
> | Quantization | VSR | DreamSim | Pruning | VSR | DreamSim |
> |---|---|---|---|---|---|
> | W8A32 | 1.000 | 0.126 | 30% | 1.000 | 0.120 |
> | W8A8 | 1.000 | 0.143 | 50% | 1.000 | 0.145 |
> | W4A32 | 0.091 | 0.174 | 70% | 0.782 | 0.168 |
>
> - **No additional constraints.** The private classifier and reference generator do not interact with the suspect model's deployment. Replacing them with SDXL/Z-Image preserves T@10⁻⁶F and VSR at 1.000 (see Reviewer jbqZ W5).
>
> **Response to Q3: LFS Consistency Across Fine-Tuning Strategies**
>
> Thank you for this valuable question!
>
> - **Empirical effectiveness.** Tab. 3/5 show near-perfect T@10⁻⁶F under LoRA and DreamBooth. Under DreamBooth, top-3 sensitive layers remain in normalization/cross-attention, bottom-3 in time-embedding/bias, **closely matching Tab. 8**.
> - **Why consistent.** Normalization and cross-attention layers mediate feature adaptation and semantic conditioning, inherently sensitive to domain shift. Time-embedding layers parameterize timesteps with limited content dependence. **LoRA** only updates attention layers, already in the high-LFS region.
> - **Parameter-level granularity** would substantially increase cost. Layer-level allocation yields strong performance (Tab. 2–7). We will explore finer-grained strategies in future work.
>
> [1] Zhao et al. "A Recipe for Watermarking Diffusion Models." arXiv, 2023.
> [2] Wang et al. "SleeperMark." CVPR, 2025.

---

> > ### Author Rebuttal · Reviewer_4kjn · 2026-04-07
> >
> > Thank you for the responses. The experiments largely address my concerns. However, all key supplementary content remains only in the rebuttal and has not been integrated into the main text. I believe the content on cross-architecture generalization should be substantially expanded and incorporated into the main paper, as it currently exists only in condensed form within the rebuttal.

---

> > > ### Author Response · Authors · 2026-04-07
> > >
> > > Thank you very much for your thoughtful follow-up and for acknowledging the value of our additional experiments. We are truly glad that these results have largely addressed your concerns.
> > >
> > > We completely agree with you that the cross‑architecture generalization results are critical and would substantially strengthen our work. We sincerely apologize that these contents currently remain only in the rebuttal. However, as you may understand, during the rebuttal period we are not allowed to directly modify the submission PDF.  We believe that the purpose of the rebuttal is exactly to provide a channel for us to address reviewers' concerns, typically through additional experiments and clarifications, before the final revision.
> > >
> > > Given this, we would like to reassure you that in the final revision we will substantially expand the cross‑architecture generalization section, integrate all relevant results and discussions into the main paper, and provide more thorough analyses. We truly appreciate your constructive suggestion, which will help us improve the paper significantly.
> > >
> > > We hope that our commitment to fully incorporating these points in the revision can address your remaining concern, and we would be very grateful if you could consider raising your score accordingly. Thank you again for your time and valuable feedback.

---

### Decision · Program_Chairs · 2026-04-30

**Decision:**

Accept (regular)

**Comment:**

In this paper, the authors address the critical issue of the vulnerability of intellectual property and model ownership verification in text-to-image diffusion models. They proposed Cert-LAS that is claimed to be the first certified MOV method for T2I models based on layer-adaptive smoothing.
After the rebuttal process, all the four reviewers have been satisfied with the authors' responses, consistently recognize the contributions of this paper, including (1) Rigorous mathematical framework by extending randomized smoothing techniques to the parameter space of high-dimensional generative models; (2) Cert-LAS provides robustness guarantees under certain-conditions; (3) The paper is original, practically relevant, and promising, and give positive scores (>= 4).
Despite the overall positive scores, some concerns from the reviewers are not full resolved, and are treated the limitations in some aspect of this work.
Based on the above concerns, this paper is recommended to be accepted with reservations!